# Heat Transfer in 3D Laguerre–Voronoi Open-Cell Foams under Pulsating Flow

Aidar Khairullin [1,*], Aigul Haibullina [1,*], Alex Sinyavin [1], Denis Balzamov [1], Vladimir Ilyin [1], Liliya Khairullina [2] and Veronika Bronskaya [2,3]

1   Energy Supply of Enterprises, Construction of Buildings and Structures, Kazan State Power Engineering University, 51 Krasnoselskaya Street, 420066 Kazan, Russia
2   Engineering Institute of Computer Mathematics and Information Technologies, Kazan Federal University, 18 Kremlyovskaya Street, 420008 Kazan, Russia
3   Mechanical Faculty, Kazan National Research Technological University, 68 Karl Marx Street, 420015 Kazan, Russia
*   Correspondence: kharullin@yandex.ru (A.K.); haybullina.87@mail.ru (A.H.)

**Abstract:** Open-cell foams are attractive for heat transfer enhancement in many engineering applications. Forced pulsations can lead to additional heat transfer enhancement in porous media. Studies of heat transfer in open-cell foams under forced pulsation conditions are limited. Therefore, in this work, the possibility of heat transfer enhancement in porous media with flow pulsations is studied by a numerical simulation. To generate the 3D open-cell foams, the Laguerre–Voronoi tessellation method was used. The foam porosity was 0.743, 0.864, and 0.954. The Reynolds numbers ranged from 10 to 55, and the products of the relative amplitude and the Strouhal numbers ranged from 0.114 to 0.344. Heat transfer was studied under the conditions of symmetric and asymmetric pulsations. The results of numerical simulation showed that an increase in the amplitude of pulsations led to an augmentation of heat transfer for all studied porosities. The maximum intensification of heat transfer was 43%. Symmetric pulsations were more efficient than asymmetric pulsations, with Reynolds numbers less than 25. The Thermal Performance Factor was always higher for asymmetric pulsations, due to the friction factor for symmetrical pulsations being much higher than for asymmetric pulsations. Based on the results of a numerical simulation, empirical correlations were obtained to predict the heat transfer intensification in porous media for a steady and pulsating flow.

**Keywords:** pulsating flow; heat transfer enhancement; porous media; Laguerre–Voronoi tessellation; open-cell foams; heat transfer coefficient





## 1. Introduction

Open-cell metal foams are highly porous materials with randomly arranged cell structures. Metal foams have high thermal conductivity, high specific surface area, and high porosity. Metal foams have a complex three-dimensional structure, which enhances convective heat transfer due to flow mixing. Porous materials have interesting characteristics that make them attractive for many engineering applications, such as fuel-cell stacks [1] compact heat exchangers [2,3], two-phase gas–liquid mixtures [4], reactors [5], filters [6], and other heat exchange applications [7,8].

Although there are more theoretical than experimental studies of the heat transfer and flow characteristics of metal foams of porous structures [9], the number of works in this latter area continues to grow [10–13]. The heat transfer and flow characteristics in porous structures are mainly investigated in a steady flow, while studies with forced unsteady flows are limited. A forced flow pulsations is one of the methods of heat transfer intensification. The effectiveness of pulsations for heat transfer enhancement has been shown in many studies [14,15]. Therefore, the combined use of porous structures under the conditions of the pulsating flow to intensify heat transfer is an interesting problem.

Ni et al. [16] studied the characteristics of oscillating flow on wire screens based on a combined numerical and experimental study. The frequency was from 2 to 7.9 Hz. Significant phase shifts between piston movement and gas displacement within the porous medium were found, the phase difference increasing with increasing pressure loss. A correlation has been proposed to allow the prediction of the phase difference. Leong et al. [17] conducted experimental studies of heat transfer and fluid flow in metal foams with oscillating flow. Leong et al. [17] noted that the characteristics of the flow significantly depended on the kinetic Reynolds number based on the oscillatory frequency and the displacement amplitude of the flow. The heat transfer and pressure loss in the metal foam increased with the growth of the kinetic Reynolds number and the displacement amplitude of the flow. The influence of the kinetic Reynolds number is more significant on pressure losses compared to the dimensionless pulsation amplitude. The authors of [18] experimentally studied heat transfer in aluminum foam with an oscillating flow. It was shown that with an increase in the kinetic Reynolds number from 178 to 874 for dimensionless amplitude of 3.1–4.1, the Nusselt number increases. The authors in [19] presented the experimental results of the heat transfer characteristics of metal foam heat sinks of different pore densities in an oscillating flow. According to the authors [19], aluminum foams have better heat transfer characteristics than finned heat sinks. The heat transfer rate of aluminum foam increases with the increase in the pore per inch (PPI). At the same pumping power, foams with lower PPIs have better heat transfer characteristics. Kahaleras et al. [20] presented experimental results on the flow characteristics of different metallic regenerators under alternating air flow. The authors found that with an increase in the kinetic Reynolds number, the friction factor for a 30% porosity regenerator decreases. For regenerators with porosities of 35% and 45%, the kinetic Reynolds number has almost no effect on the friction factor. Dellali et al. [21] presented the results of an experimental study in a porous regenerator under conditions of an oscillating flow. The paper shows that pressure drop depends on frequency, amplitude, and porosity. The coefficient of friction in an oscillating flow is higher than the steady flow for the highest porosity; at lower porosities, the friction coefficients are the same. Bayomy et al. [22] presented the results of a numerical and experimental study in aluminum foam under a pulsating water flow with a frequency ranging from 0.04 to 0.1 Hz. The results showed an increase in heat transfer by 14% in the pulsating flow compared to the steady flow. The authors conclude that a pulsating flow leads to a more uniform temperature distribution compared to a steady flow. The authors of [23–25] experimentally studied the oscillating water flow in aluminum foam from 10 to 40 PPI, with a porosity of 88% a pulsation frequency of 0.116 to 0.696 Hz, and flow displacement amplitudes from 74.35 to 111.53 mm. According to the authors, PPI had a significant impact on such parameters as the friction factor, pressure drop, and inlet pressure. The authors have shown that pressure losses increase both with increasing frequency and the amplitude of pulsations. The friction factor was higher in oscillating flow compared to steady flow. At high frequencies, the maximum inlet pressure was recorded for foam with 20 PPI, and at minimum frequencies foam with 10 PPI showed the highest inlet pressure. A phase difference was recorded between inlet and outlet pressure and flow rate. To reduce computational costs, in the numerical study of porous media under the conditions of pulsating flows, a simplified two-dimensional geometry was used [26–28]. Kim et al. [26] numerically studied pulsating flows in a porous structure represented as two-dimensional arrays of square cylinders. The porosity was in the range from 0.64 to 0.84, and the pulsation frequency was from 20 to 64 Hz. The authors concluded the porosity affected permeability and the frequency strongly affected Forchheimer coefficients. The authors in [27] studied oscillating flows in a porous medium numerically using the Lattice Boltzmann Method. The authors concluded that the temperature and velocity profiles were affected by the Womersley number. Chen et al. [28] studied heat transfer and hydrodynamics in a porous medium with a two-dimensional numerical simulation. Arrays of cylinders represented the porous medium. The results showed that the Nusselt number in the pulsating flow differed significantly from the steady flow. Heat transfer enhancement strongly depended on the

pulsating frequency and amplitude of pulsation. To simplify the numerical simulation in a porous medium, Ghafarian et al. [29] used the Darcy–Brinkman–Forchheimer model. The authors in [29] found that a channel with a porous medium under the conditions of an oscillating air flow leads to a significant heat transfer augmentation. The high amplitude and frequency of the pulsations leads to an increase in the local Nusselt number. Recently, Habib et al. [30] used numerical simulations to investigate the convective response in a porous medium as a function of a pulsating input flow. The porous medium was presented as an array of tubes. The authors show that as the Reynolds number increases, the nonlinearity between the Nusselt number and the flow fluctuations at the inlet increases. Further increase can restore the linearity response.

In the numerical simulation of the porous media, it is important to create a complex geometry of a porous structure. Various methods are used to create this geometry [31–33]. Iasiello et al. and Diani et al. [34,35] employed X-ray micro-tomography to reproduce the geometry of the porous medium. The authors in [34,35] showed that the constructed porous medium predicts the characteristics of heat transfer and pressure drop in the porous medium, with good agreement with experimental data. Despite the problems with accurately representing the proper structure, micro-tomography allows the reproduction of the complex irregular structure of a porous medium. In addition, this method requires expensive equipment. The construction of the geometry of a porous structure is often performed using regular idealized Kelvin cell structures [36–38] or Weaire and Phelan cells [39]. Some authors use randomized Kelvin structures [40] or Kelvin structures with elliptical struts [41]. The body-centered-cubic [42–44] and cubic cell model [45] are also used to reproduce idealized structures. Another method for creating a realistic geometry of a porous medium is based on Laguerre–Voronoi tessellation (LVT) [46]. This method gives good agreement with the experimental data [47]. For flow simulation in a porous medium, the LVT technique is used less often than other methods, which is associated with the complexity of this method. Some authors construct the geometry of a porous medium by numerically duplicating the formation process of foam [48], which is in a good agreement when comparing the specific surface area of constructed foam with real foam.

Studies of the pulsating flows in porous structures are limited. Experimental studies with the oscillating flow in porous structures are carried out without unidirectional flow conditions [16–21,23–25]. Such flows cannot be directly compared with a steady flow. The results are difficult to apply to pulsating flows with unidirectional flow conditions. Additionally, in many experimental works, only the flow characteristics are studied, without the heat transfer [16,17,21,23–25]. Numerical simulation using the realistic 3D geometry of a porous medium can be difficult even for a steady flow (for large computational domains). Therefore, the numerical simulation of pulsating flows is performed with two-dimensional models [26–28] or using other simplifications [29]. In all the studies mentioned in the review, the flow pulsations are symmetrical. The pulsations can be both symmetric and asymmetric [49]. The heat transfer enhancement with asymmetric pulsations can exceed symmetrical pulsations. The better thermal performance for asymmetrical pulsation compared to symmetrical pulsations for an impinging air jet is shown in [50], and for a tube bundle in cross flow [51]. In our previous work [52], limited results were presented for a porosity of 0.954, pulsation amplitude of 28.6, and Reynolds number of 30 on Voronoi foam (VF).

This study aims to investigate the heat transfer and friction factor, and obtain an empirical correlation in LVT generated foams, under symmetrical and asymmetrical air flow pulsations with unidirectional flow conditions. Studies with pulsating flows in VF are extremely limited. This article presents the results of thermal performance in Voronoi foam for Reynolds numbers 10, 25, 40, and 55, product of relative amplitude and Strouhal numbers 0.114, 0.191, 0.268, and 0.344, duty cycle of pulsation 0.25 and 0.5, and porosity 0.743, 0.864, and 0.954.

## 2. Numerical Simulation

### 2.1. Voronoi Foam Generation

The construction of the 3D foam structure was based on Laguerre–Voronoi tessellation in four steps. In the first step, a random filling of densely spaced spheres was generated in MFIX-DEM [53]. The required pore density was set by the diameter of the spheres.

In the second step, based on sphere center's coordinates, the LVT was performed using the free software Voro++ [54]. As a result, two output files contained 3D space coordinates of cell vertexes and a vertex order of faces were obtained.

In the third step, the vertex coordinates of each cell strut were determined by processing the second step output files using Excel VBA.

In the fourth step, the surface geometry was constructed. The final geometry was created using the Ansys SpaceClaim addition programmed in the VisualBasic.Net. Figure 1 shows the foam generated by the LVT method.

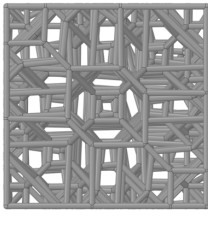

(**a**)

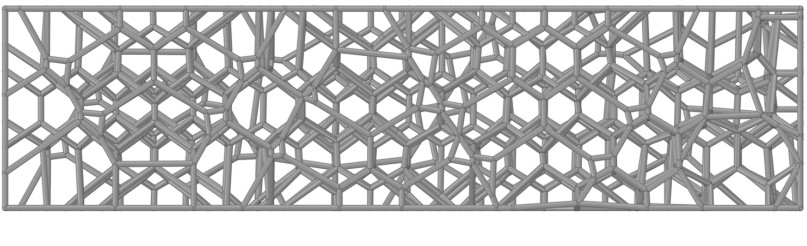

(**b**)

**Figure 1.** Generated VF: (**a**) Front view; (**b**) Right view.

### 2.2. Governing Equations and Boundary Conditions

For the aims of the numerical study, three VF with porosities of 0.743, 0.864, and 0.954 were generated. The required porosity of the VF was achieved by changing the struts' diameters. The foam struts were cylindrical. All generated foams had a cell diameter of 25.4 mm and PPI of 2. To exclude the wall effect, only the central region of the generated VF (Figure 1) was used in the numerical simulation. The computational domain of the VF is shown in Figure 2. The geometric characteristics of the generated VF are summarized in Table 1.

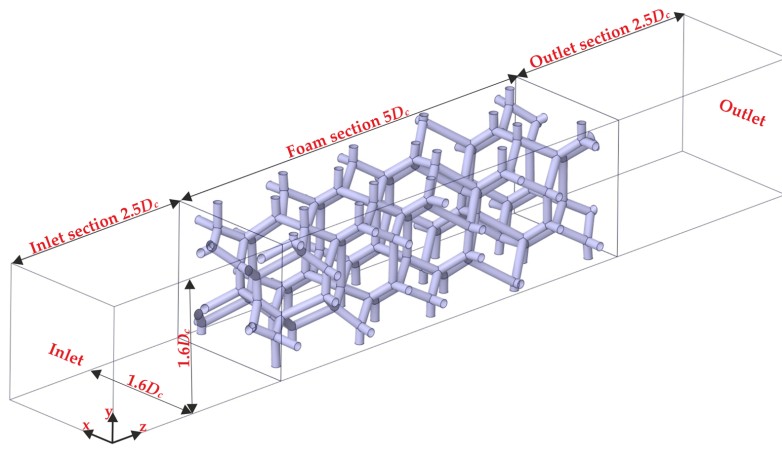

**Figure 2.** Computational domain of the VF.

**Table 1.** Geometrical characteristics of the generated VF.

| Tested VF | PPI [in$^{-1}$] | $\varepsilon$ [–] | $D_s$ [mm] | $D_c$ [mm] | $a_{sv}$ [m$^2$ m$^{-3}$] |
|---|---|---|---|---|---|
| VF-0.743 | 2 | 0.743 | 6.5 | 25.4 | 145 |
| VF-0.864 | 2 | 0.864 | 4.5 | 25.4 | 115 |
| VF-0.954 | 2 | 0.954 | 2.5 | 25.4 | 72 |

The pore number in the flow direction affect the results of numerical simulation. Numerical simulation results in open-cell foams affected by the pores number in the flow direction. However, with an increase in the size of the computational domain, it is necessary to use a larger number of grid elements, which increases the calculation time and requires more computational power. Diani et al. [35] showed that 10 pores in the flow direction are sufficient for a fully developed flow in open-cell foam. The authors in [47] also used 10 pores in the flow direction with VF. Therefore, the number of pores in the flow direction was 10. The length of the input and output buffer zones of the computational domain was 5 pores.

The mass, momentum, and energy equations were solved directly with the finite volume method [35]. The fluid flow was assumed to be incompressible. The symmetry condition was set on the sides of the computational domain. A constant heat flux of 5 W/m², with no slip condition, was set on the foam walls. At the inlet to the computational domain, the constant temperature was 299.15 K. At the steady flow at the inlet of the computational domain, a constant flow velocity was set, with an outlet pressure of 101,325 Pa. Velocity pulsations were set at the inlet for the pulsating flow. The pulsating velocity corresponded to the required Reynolds number, frequency, amplitude, and duty cycle of pulsation. The flow pulsations had a reciprocating character. The shape of the velocity pulsations for symmetric and asymmetric pulsations is shown in Figure 3. The pulsation system with a similar velocity pulsation shape is shown in our previous work [55].

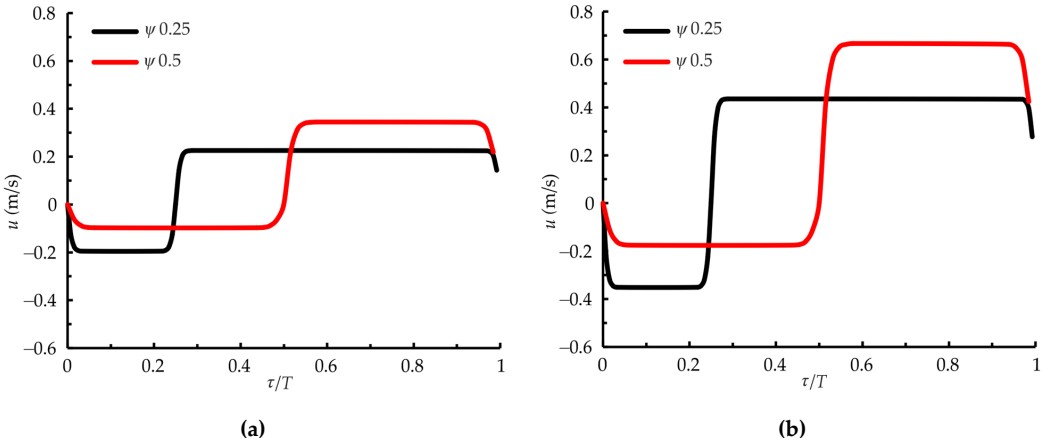

(a)　　　　　　　　　　　　　　　　　　　　　(b)

**Figure 3.** The pulsating velocity examples at the inlet of the computational domain: (**a**) $\varepsilon = 0.743$, $Re = 40$, $(A/D_s)St = 0.344$. (**b**) $\varepsilon = 0.954$, $Re = 25$, $(A/D_s)St = 0.344$.

The frequency of pulsation is defined as:

$$f = 1/T, \tag{1}$$

where $T$ is the pulsation period:

$$T = T_1 + T_2,$$

where $T_1$ is the first half-period of pulsation which corresponded to the reverse flow in the foam; $T_2$ is the second half-period of pulsation. The duty cycle of pulsation is found by the equation:

$$\psi = T_1/T. \tag{2}$$

The amplitude of pulsation is defined as:

$$A = \frac{\int_0^{T_1} u \, d\tau}{T_1}, \tag{3}$$

where $u$ is the instantaneous velocity of the flow at the inlet of the computational domain.

The Reynolds number based on strut diameter $D_s$ was found by the equation:

$$Re = \frac{\rho u_{st} D_s}{\mu \varepsilon},$$

(4)

where $u_{st}$ is the steady velocity of the flow at the inlet of computational domain, and $\rho$ and $\mu$ are the density and dynamic viscosity of the air, respectively. The steady velocity of the flow $u_{st}$ is equal to the pulsation velocity up averaged over the one period of the pulsation:

$$u_{st} = \langle u_p \rangle = \frac{\int_0^T u d\tau}{T}.$$

(5)

The Strouhal number is found by equation:

$$St = \frac{f D_s \varepsilon}{u_{st}}.$$

(6)

The Nusselt number is defined as:

$$Nu = \frac{q D_s}{(t_w - t_{air}) k},$$

(7)

where $q$ is the heat flux imposed at the foam wall, $t_w$ is the mean wall temperature of the foam, $t_{air}$ is the bulk mean air temperature, and $k$ is the thermal conductivity of air. The bulk mean air temperature was averaged for foam section (Figure 2).

The Nusselt number averaged over one period of pulsation is found by the equation:

$$\langle Nu_p \rangle = \frac{\int_0^T Nu_p d\tau}{T}.$$

(8)

The friction factor averaged over one period of pulsation is defined as:

$$\langle \xi_p \rangle = \frac{\langle \Delta P u \rangle 2}{\rho u_{st}^3},$$

(9)

where $\Delta P$ is the pressure drop in VF.

*2.3. Detail of Solution Methodology*

The numerical study was performed at Reynolds numbers *Re* 10, 25, 40, and 50. Since the Reynolds number was based on the strut diameter, the flow velocity changed for each generated VF. The working medium was air. The heat capacity, thermal conductivity, dynamic viscosity, and density of the air were 1005 J/(kg K), 0.026 W/(m K), $1.844 \times 10^{-5}$ Pa s, and 1.182 kg/m$^3$, respectively. The Prandtl number *Pr* corresponded to 0.713. Three VF were generated with porosities $\varepsilon$ of 0.743, 0.864 and 0.954, and strut diameters *Ds* of 6.5, 4.5 and 2.5 mm, respectively.

Pulsating flow is mainly characterized by the amplitude and frequency of pulsations. However, in this study, the influence of the frequency and amplitude of pulsations is not studied separately. Unsteady simulation in 3D porous structures requires significant computational resources. A separate study of the frequency and amplitude of pulsations increases the number of simulation cases. To reduce the solving time, the influence of the product of the dimensionless pulsation amplitude and the Strouhal number $(A/D_s)St$ was studied.

The dimensionless group $(A/D_s)St$ was performed at 0.114, 0.191, 0.268, and 0.344. All calculations were performed at the pulsation frequency 2 Hz. Strouhal was different depending on the Reynolds number. The required values of the dimensionless group

$(A/D_s)St$ were selected by changing the pulsation amplitude. The Strouhal number at various Reynolds numbers and the strut diameters ranged from $1.456 \times 10^{-2}$ to $54.14 \times 10^{-2}$. The amplitude of the pulsations was small enough so that the reverse flow from the outlet did not reach the foam section. During the calculation, the pulsations' amplitude $A/D_s$ ranged from 0.211 to 23.59.

All calculations were performed for symmetric and asymmetric flow pulsations. The effect of asymmetrical pulsation on VF thermal performance was investigated, with a duty cycle of 0.25 and 0.5 for asymmetric and symmetric flow pulsations, respectively. For a pulsating flow, 96 separate cases of calculation were carried out, and for a steady flow, 12. Numerical simulation was performed in Ansys Fluent. The pressure-based coupled solution algorithm was employed to solve incompressible Navier-Stokes equations. For all simulation cases, the PISO algorithm was used, which is more suitable for unsteady flow. The residual of $10^{-4}$ was used for mass and momentum equations and the residual of $10^{-6}$ was used for energy equations. For time stepping, the bounded second order transient formulation was used. Time step was $10^{-4}$. For a pulsating flow, the solution marched until reaching a quasi-steady regime. The quasi-steady regime was reached in 4–8 pulsation periods, depending on the pulsation regime. The solution of a steady flow was applied as the initial conditions. A quasi-steady regime is considered to be when the Nusselt values averaged over the period of pulsations and pressure drop between two adjacent periods differed by less than 0.2%.

### 2.4. Grid Independency Test and Verification of Mathematical Model

Seven polyhexo meshes were generated using Ansys Meshing for the grid independency test. The number of grid elements was 760,895 (G1), 1,229,550 (G2), 1,676,862 (G3), 3,418,269 (G4), 5,826,968 (G5), and 8,884,271 (G6). The grid independency tests were performed at Reynolds numbers 10, 25, 40, and 55, and with a porosity of 0.954. Figure 4 shows the influence of the grid density on the Nusselt number. Cases with higher Reynolds numbers are more sensitive to increased grid elements. When the Reynolds number was 10, the Nusselt number of grids G1 and G2 changed by 0.41%, and for the Reynolds number 55 by 3.01%. The maximum deviation of the Nusselt number for the entire range of Reynolds numbers for the grids G4, G5, and G6 were 0.98%, 0.31% and 0.27%, respectively. To reduce calculation time, the G4 mesh with 3,418,269 elements was chosen (Figure 5) for numerical simulation. For the G4 mesh, the ratio of the maximum mesh element size to cell diameter $y_{max}/D_c$ was 0.022. The ratio $y_{max}/D_c$ 0.022 was chosen for other porosities, as well. The number of grid elements was less for other porosities, as a result of the decrease in porosity. For porosities of 0.864 and 0.743, the number of elements was 2,094,495 and 1,787,001, respectively. The boundary layer contained six layers. The cell size of the first three layers expanded in the radial direction with a factor of 1.2.

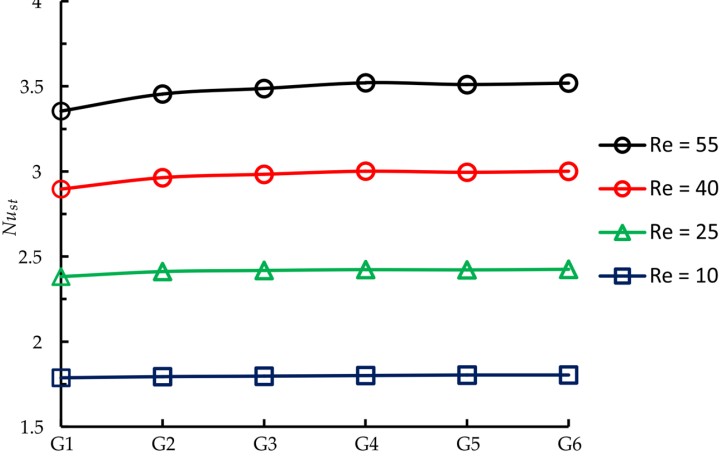

**Figure 4.** The grid independency test.

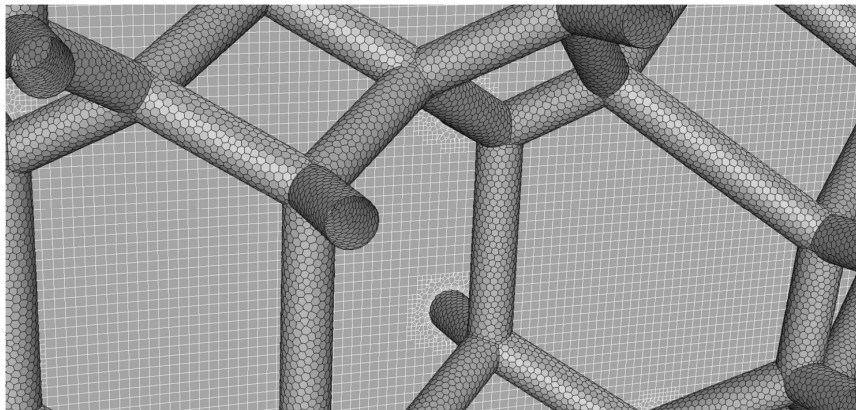

**Figure 5.** The Voronoi foam grid (G4).

To verify the numerical simulation, the Nusselt number obtained for the steady flow at a porosity of 0.954 was compared to the experimental data of other authors. Figure 6 shows the Nusselt number obtained from the numerical simulations and empirical correlations proposed by Mancin et al. [56] and Calmidi et al. [57]. As can be seen from Figure 6, there is a good agreement between the Nusselt numbers and the data from Mancin et al. [56] and Calmidi et al. [57]. The difference between numerical simulation and experimental data from [56,57] was less than 9%.

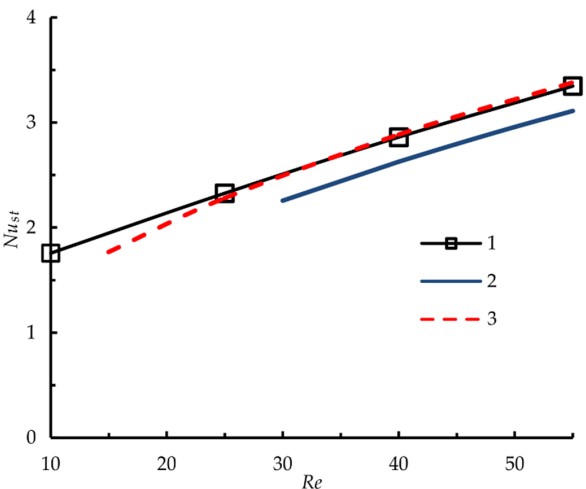

**Figure 6.** Verification of numerical simulation: 1 – Present study; 2 – Mancin et al. [56]; 3 – Calmidi et al. [57].

## 3. Results and Discussion

The condition parameters of the Reynolds and Prandtl numbers in this study were chosen to correspond electronic cooling applications. The pulsating condition parameters were varied to investigate the effect on thermal performance VF. The maximum values of the dimensionless group $(A/D_s)St$ were chosen to consider the cost of numerical simulation. Since some studies have shown the effectiveness of asymmetrical pulsations compared with symmetrical ones [50,51], both types of pulsations were considered in this study.

### 3.1. The Velocity Streamlines and Contours Plots

Figures 7 and 8 show the velocity contours and streamlines for a porosity of 0.743 and 0.954 and for a steady flow at a Reynolds number of 25. The VF struts behaved as an obstacle to the flow, and stagnant zones with a reduced velocity formed behind the VF ligaments. As the porosity decreases to 0.743 (Figure 7b), some of the VF ligaments joined, resulting in an increased stagnation zone. A decrease in the distance between the ligaments led to an increase in local velocities and a more intense mixing of the flow, which

is more clearly seen in Figure 8a,b. Despite the three-dimensional structure of the flow, at a porosity of 0.954 (Figure 8a), the streamlines were less disturbed compared to the flow in a VF with a porosity of 0.743. The nature of the flow in the VF obtained in this study was consistent with the data of other researchers. Diani et al. [35] and Nie et al. [47] used X-ray micro-tomography and LVT for the generation of the surface geometry of the foam, they a observed three-dimensional flow structure with local stagnant zones also. The flow for the cut plane in the middle section (Figure 7a,b) had a symmetrical flow pattern. At the same Reynolds number, with a porosity of 0.743 (Figures 7b and 8b), the input velocity decreased, and since the diameter of the struts increased, the input velocity decreased according to Formula (4).

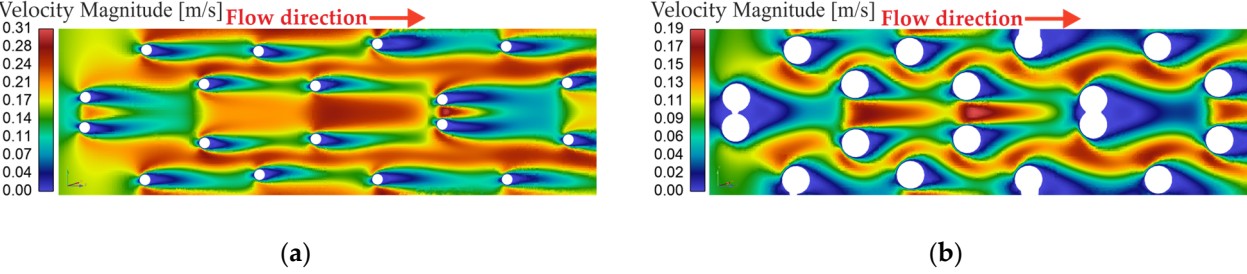

**Figure 7.** A cut plane of the velocity contours plots in middle cross section for steady flow at *Re* = 25. Unit is m/s. (**a**) $\varepsilon$ = 0.954. (**b**) $\varepsilon$ = 0.743.

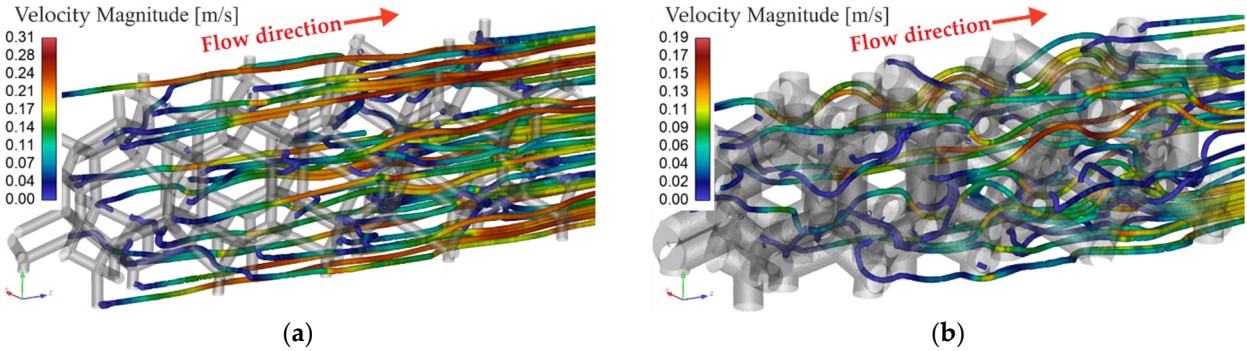

**Figure 8.** The velocity streamlines for steady flow at *Re* = 25. Unit is m/s. (**a**) $\varepsilon$ = 0.954. (**b**) $\varepsilon$ = 0.743.

Figures 9 and 10 show the velocity contours plots for different phases of the pulsating flow, for symmetric and asymmetric pulsations, and with various porosities. Velocity contours plots were obtained with a Reynolds number of 25 and dimensionless group $(A/D_s)St$ of 0.344. The shape of the velocity pulsations, which was set at the input of the computational domain for the symmetric and asymmetric pulsations, is shown in Figure 3.

The pulsating flow was more disturbed compared to the steady flow. As seen in Figures 9 and 10, for the symmetric and asymmetric pulsations, there was differences in the flow structure. The difference in the flow structure was more significant for the first half-period of pulsations, during which the flow reversed. The flow velocity for the first half-period of pulsations was less for the symmetrical pulsations, and for the second half-period of pulsations for asymmetric pulsations. For the pulsation phase $\tau/T$ = 0 (Figures 9a,b and 10a,b), the flow for the symmetrical and asymmetrical pulsations had a different flow pattern. If with the asymmetric pulsations $\tau/T$ = 0 (Figures 9a and 10a) there was some flow symmetry in the VF cells, then with the symmetrical pulsations (Figures 9b and 10b), the flow structure was more disturbed. The greater mixing of the flow structure with symmetrical pulsations for the phase $\tau/T$ = 0 was because the flow reversal occurred at higher velocities. For the pulsation phase $\tau/T$ = 0.1, the flow was changing direction. Behind the struts, stagnant zones were observed (Figures 9c,d and 10c,d), as in the steady flow (Figure 7). At the pulsation phase $\tau/T$ = 0.2, the flow is still had opposite directions for both types of pulsations (Figures 9e,f, and 10e,f), while the flow

became more structured. For the next pulsation phase $\tau/T = 0.3$ for the asymmetric pulsations, the flow was changing direction (Figures 9g and 10g). For the pulsation phases $\tau/T = 0.3$ and 0.4, symmetric and asymmetric pulsations had different flow directions. At pulsation phases $\tau/T > 0.3$ with the asymmetric pulsations, the flow structure was stabilized (Figures 9i,k,m,o, and 10i,k,m,o). With symmetric pulsations, the flow reversal occurred at the pulsation phase $\tau/T = 0.5$, which led to a complete restructuring of the flow. For pulsation phases $\tau/T > 0.5$ in the symmetric pulsations, the nature of the flow changed insignificantly, while the unsteady nature of the flow was still noticeable. The flow velocity for pulsation phases $\tau/T < 0.5$ was lower for the symmetrical pulsations, and for pulsation phases $\tau/T > 0.5$, it was lower for the asymmetric pulsations. The flow structure for the asymmetric and symmetric pulsations was close at $\tau/T > 0.5$. However, there were differences in the formation of eddies behind the struts. Due to the higher velocity for symmetrical pulsations, at $\tau/T > 0.5$, unsteady vortex separation was more noticeable. Animation of the pulsation flow shown in the Video S1.

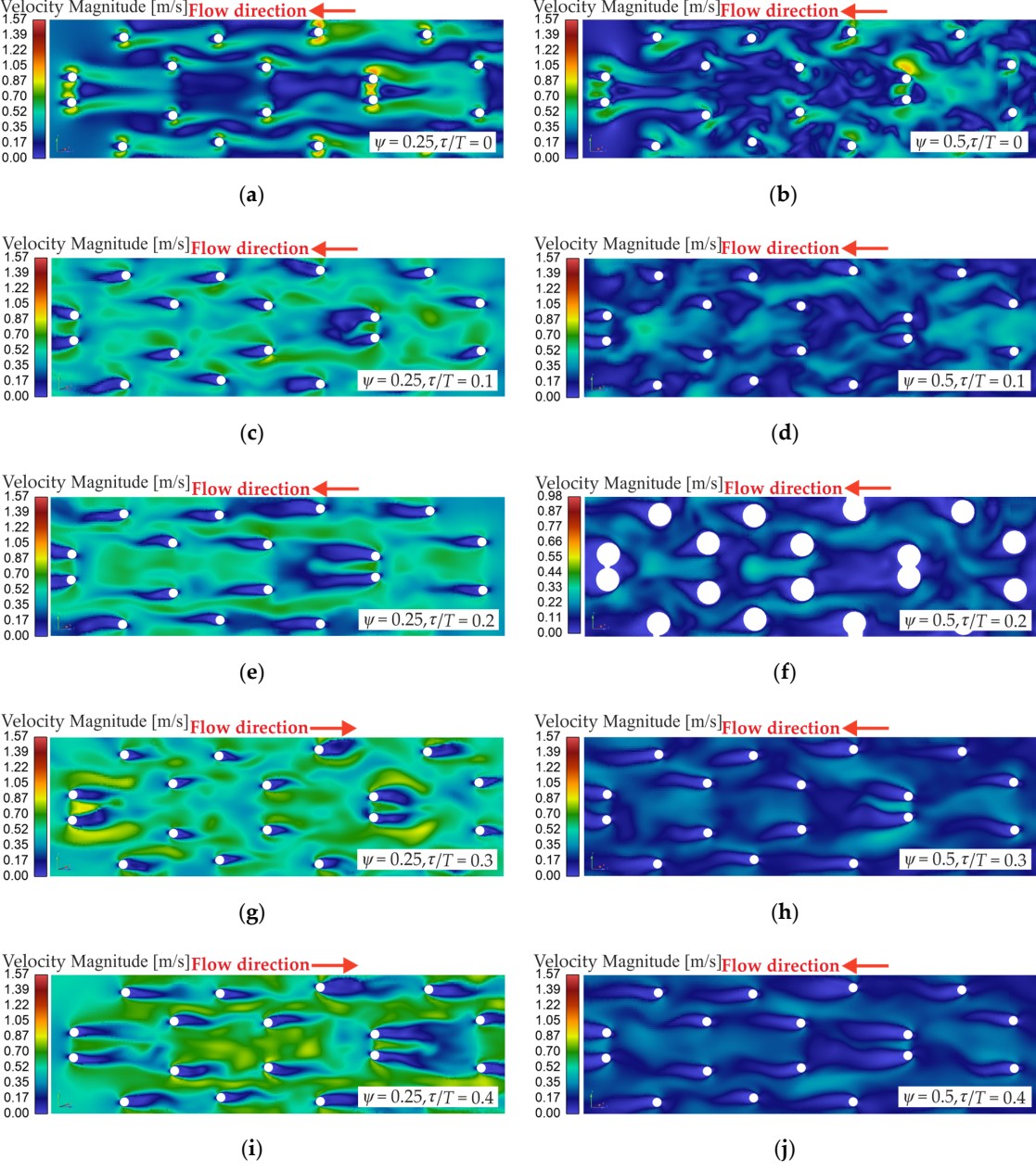

**Figure 9.** *Cont.*

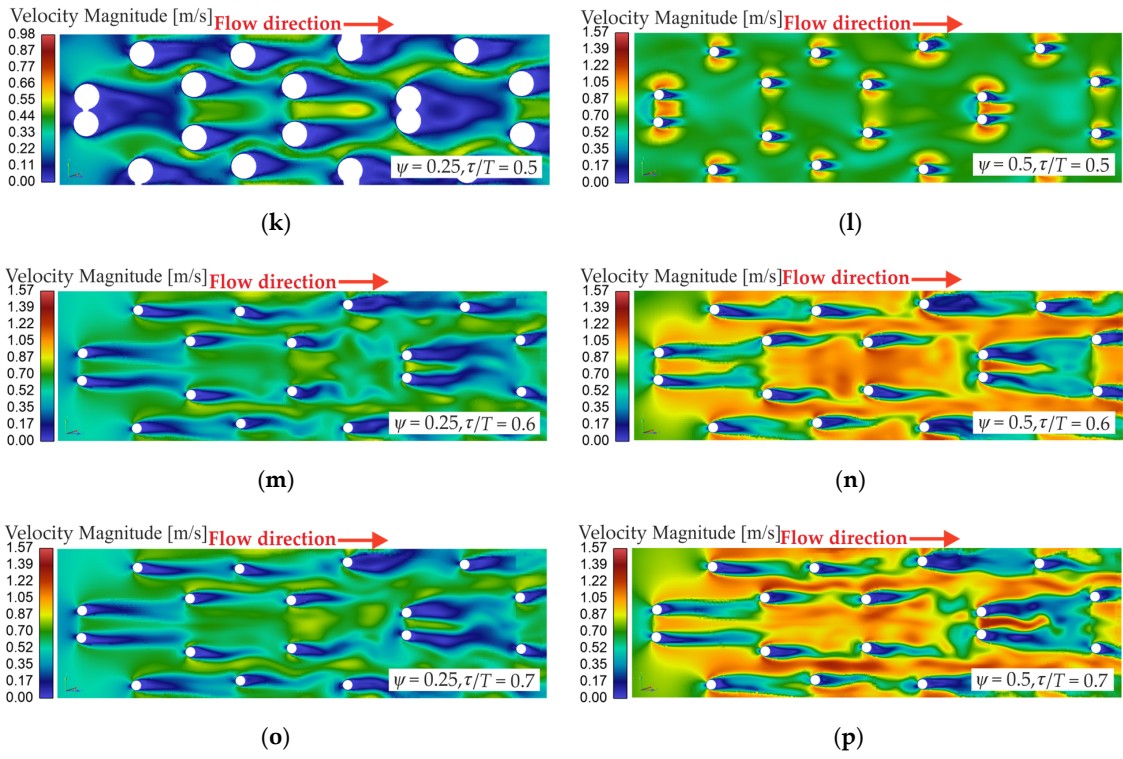

**Figure 9.** A cut plane of the instantaneous velocity contours plots in middle cross section for pulsating flow at $\varepsilon = 0.954$, $Re = 25$, $(A/D)St = 0.344$. Unit is m/s: (**a**) $\psi = 0.25$, $\tau/T = 0$; (**b**) $\psi = 0.5$, $\tau/T = 0$; (**c**) $\psi = 0.25$, $\tau/T = 0.1$; (**d**) $\psi = 0.5$, $\tau/T = 0.1$; (**e**) $\psi = 0.25$, $\tau/T = 0.2$; (**f**) $\psi = 0.5$, $\tau/T = 0.2$; (**g**) $\psi = 0.25$, $\tau/T = 0.3$; (**h**) $\psi = 0.5$, $\tau/T = 0.3$; (**i**) $\psi = 0.25$, $\tau/T = 0.4$; (**j**) $\psi = 0.5$, $\tau/T = 0.4$; (**k**) $\psi = 0.25$, $\tau/T = 0.5$; (**l**) $\psi = 0.5$, $\tau/T = 0.5$; (**m**) $\psi = 0.25$, $\tau/T = 0.6$; (**n**) $\psi = 0.5$, $\tau/T = 0.6$; (**o**) $\psi = 0.25$, $\tau/T = 0.7$; (**p**) $\psi = 0.5$, $\tau/T = 0.7$.

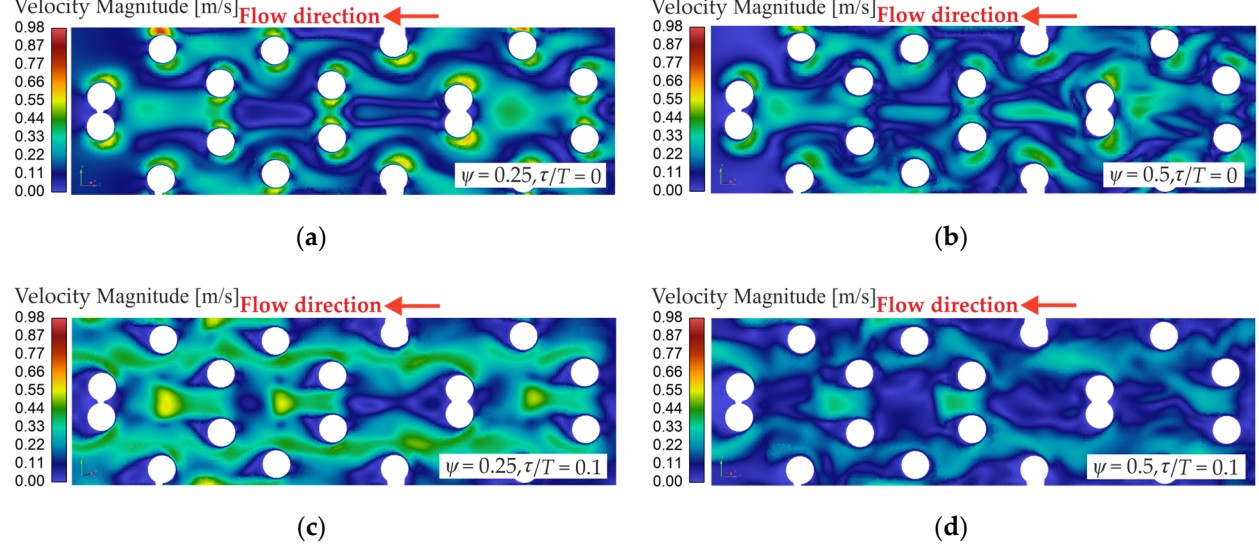

**Figure 10.** *Cont.*

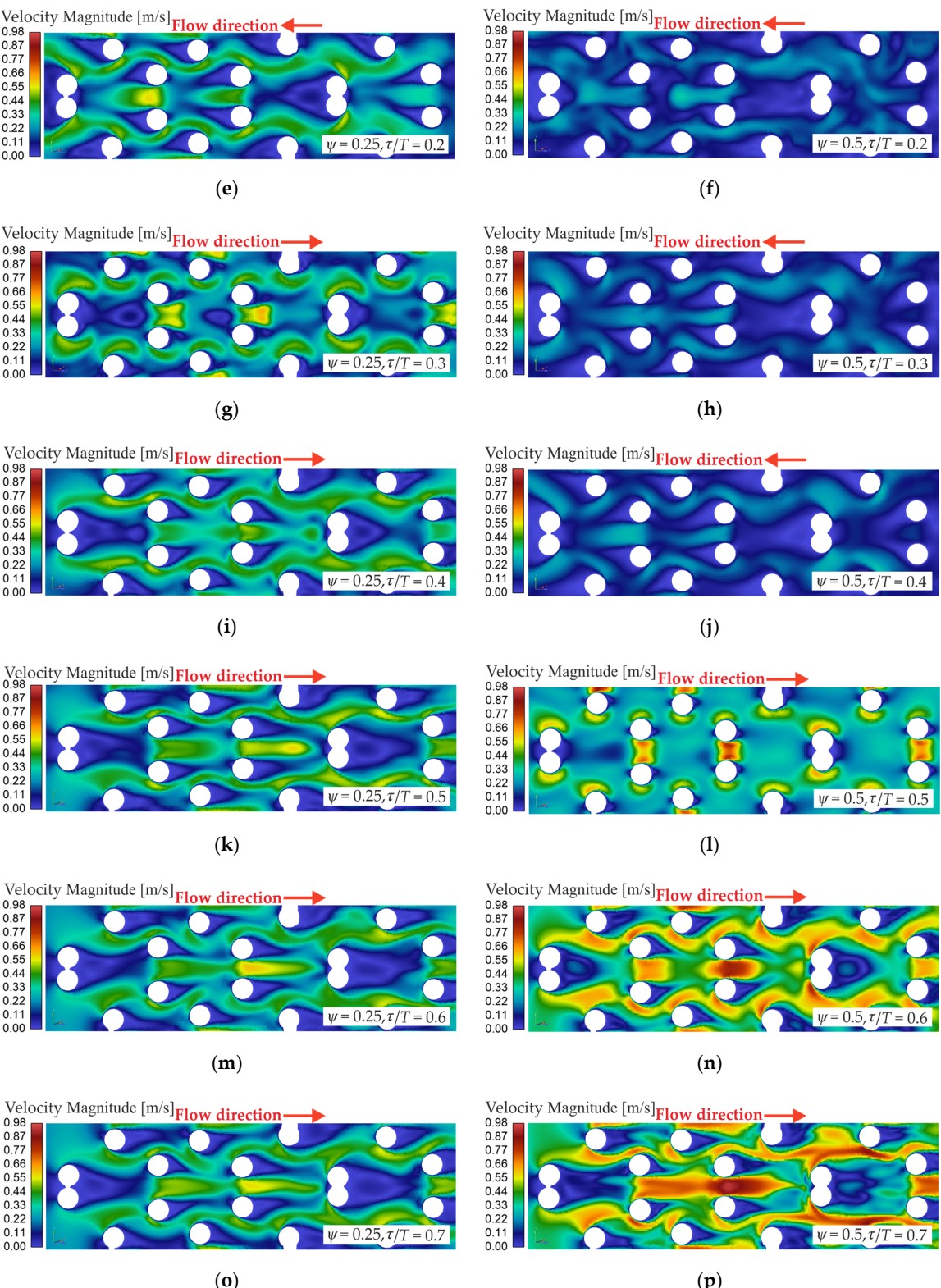

**Figure 10.** A cut plane of the instantaneous velocity contours plots in middle cross section for pulsating flow at $\varepsilon = 0.743$, $Re = 25$, $(A/D)St = 0.344$. Unit is m/s. (**a**) $\psi = 0.25$, $\tau/T = 0$. (**b**) $\psi = 0.5$, $\tau/T = 0$. (**c**) $\psi = 0.25$, $\tau/T = 0.1$. (**d**) $\psi = 0.5$, $\tau/T = 0.1$. (**e**) $\psi = 0.25$, $\tau/T = 0.2$. (**f**) $\psi = 0.5$, $\tau/T = 0.2$. (**g**) $\psi = 0.25$, $\tau/T = 0.3$. (**h**) $\psi = 0.5$, $\tau/T = 0.3$. (**i**) $\psi = 0.25$, $\tau/T = 0.4$; (**j**) $\psi = 0.5$, $\tau/T = 0.4$. (**k**) $\psi = 0.25$, $\tau/T = 0.5$. (**l**) $\psi = 0.5$, $\tau/T = 0.5$. (**m**) $\psi = 0.25$, $\tau/T = 0.6$. (**n**) $\psi = 0.5$, $\tau/T = 0.6$. (**o**) $\psi = 0.25$, $\tau/T = 0.7$. (**p**) $\psi = 0.5$, $\tau/T = 0.7$.

Figures 11 and 12 show the streamlines for the pulsation phases corresponding to reverse flow (Figures 11a,b and 12a,b) and forward flow (Figures 11c,d and 12c,d) in VF with porosities of 0.743 and 0.954. For both pulsation phases shown in Figures 11 and 12, the flow was more mixed compared with the steady flow. This was the same for the steady flow: with a decrease in porosity (Figure 12), the flow became more disturbed.

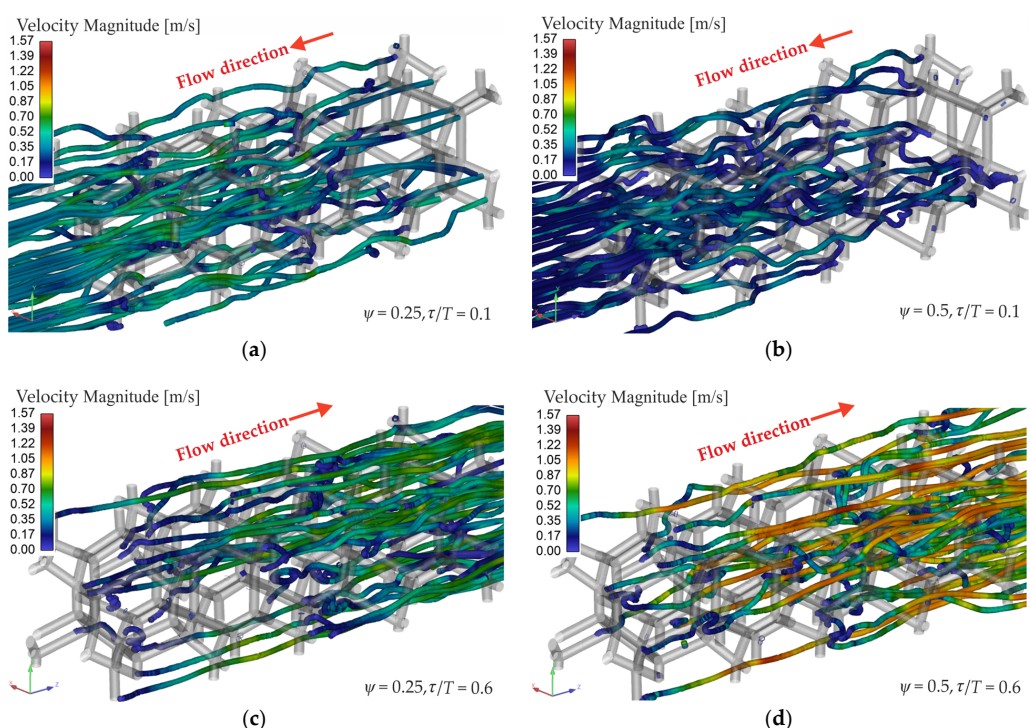

**Figure 11.** The instantaneous velocity streamlines for pulsating flow at $\varepsilon = 0.954$, $Re = 25$, $(A/D)St = 0.344$. Unit is m/s: (**a**) $\psi = 0.25$, $\tau/T = 0$; (**b**) $\psi = 0.5$, $\tau/T = 0$; (**c**) $\psi = 0.25$, $\tau/T = 0.1$; (**d**) $\psi = 0.5$, $\tau/T = 0.1$.

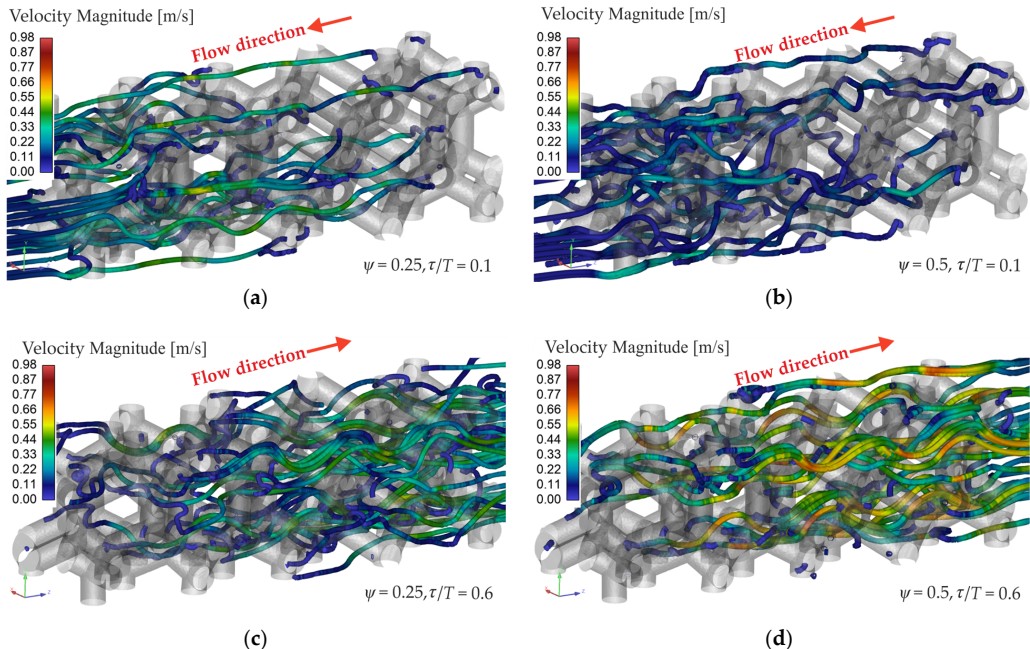

**Figure 12.** The instantaneous velocity streamlines for pulsating flow at $\varepsilon = 0.743$, $Re = 25$, $(A/D)St = 0.344$. Unit is m/s: (**a**) $\psi = 0.25$, $\tau/T = 0$; (**b**) $\psi = 0.5$, $\tau/T = 0$; (**c**) $\psi = 0.25$, $\tau/T = 0.1$; (**d**) $\psi = 0.5$, $\tau/T = 0.1$.

Due to flow pulsations, a constant restructuring of the flow occurred, which led to the equalization of velocities with a decrease in local stagnant zones. The nature of the flow was consistent with the flow with the pulsations in the tube bundles [49]. Haibullina et al. [49] studied asymmetric pulsations in an inline tube bundle at a Reynolds number of 500. Pulsating flows in the VF have some similarities with pulsating flows in tube bundles, while the flow has a three-dimensional structure with unsteady vortex separation at the lower Reynolds number.

### 3.2. The Effect of Pulsations on the Nusselt Number Ratio

The Nusselt number ratio was calculated by the formula:

$$\langle \delta Nu \rangle = \frac{\langle Nu_p \rangle}{Nu_{st}}. \tag{10}$$

Figure 13 shows the ratios of the Nusselt number averaged over one period of pulsation for the various porosities, Reynolds numbers, and duty cycles of pulsation. An increase in the intensity of pulsations $(A/D_s)St$ leads to an increase in the Nusselt number ratio, which is consistent with the data of other researchers [18,19,22,28,29,49,58]. The intensification of heat transfer is associated with a decrease in the boundary layer, an increase in local velocities, homogenization, and, additionally, more intense mixing of the flow [28,49]. Depending on the Reynolds number, the heat transfer enhancement is different. The Nusselt number ratio is generally higher at a higher Reynolds number, which is in agreement with Chen et al. [28]. Figure 13 also shows that symmetrical and asymmetrical pulsations had a different effect on the Nusselt number ratio. Sailor et al. [50] investigated the effect of the duty cycle of pulsation on heat transfer for an impinging air jet. Sailor et al. [50] showed that the best intensification of the heat transfer was observed with asymmetric pulsations. Additionally, Sailor et al. [50] noted that the effect of duty cycle was different depending on the pulsating parameters. Similar results were obtained by Ilyin et al. [51] on heat transfer in the tube bundles with pulsating flow.

When the porosity was 0.743 (Figure 13a), the heat transfer intensification was higher for the asymmetric pulsations at Reynolds numbers greater than 25, except for the conditions with a minimum pulsation intensity $(A/D_s)St$. For a Reynolds number of 10 and $(A/D_s)St$ of 0.114, the heat transfer enhancement was higher for the asymmetric pulsations; for a Reynolds number of 40 and $(A/D_s)St$ of 0.114, the heat transfer enhancement was higher for symmetric pulsations. The difference between the symmetric and asymmetric pulsations was insignificant. The symmetric and asymmetrical pulsations had a maximum difference of 4%. The shift between the two types of pulsations increased with increasing pulsation intensity at a Reynolds number of 55 and decreased at a Reynolds number of 25. At the minimum Reynolds number, the heat transfer intensification was minimal. As the Reynolds number increased, the Nusselt number ratio increased. The effect of pulsation intensity on the enhancement ratio was different depending on the Reynolds number. As the Reynolds number increased, an increase in $(A/D_s)St$ had a greater effect on the Nusselt number ratio. For example, at a Reynolds number of 10 and a duty cycle of 0.25, the Nusselt number ratio increased from 0.99 to 1.03 at $(A/D_s)St$ of 0.114 and 0.344, respectively. For a Reynolds number of 55 and a duty cycle of 0.25, the heat transfer enhancement increased from 1.15 to 1.43 at $(A/D_s)St$ 0.114 and 0.344, respectively. The maximum intensification of heat transfer was observed at the maximum Reynolds number and pulsation intensity. The maximum increase in the Nusselt number ratio was 1.43 at *Re* 55, $(A/D_s)St$ 0.344, and $\psi$ of 0.25, which is higher than the heat transfer enhancement shown by Bayomy et al. [22]. In [22], the ratio of the Nusselt number for a pulsating flow in metal foam was 1.14. For a porosity of 0.743, there were cases in which there was a slight decline in the heat transfer intensification. At the minimum Reynolds number and pulsation intensity, the Nusselt number ratio in a pulsating flow decreased by 1–2% compared with a steady flow.

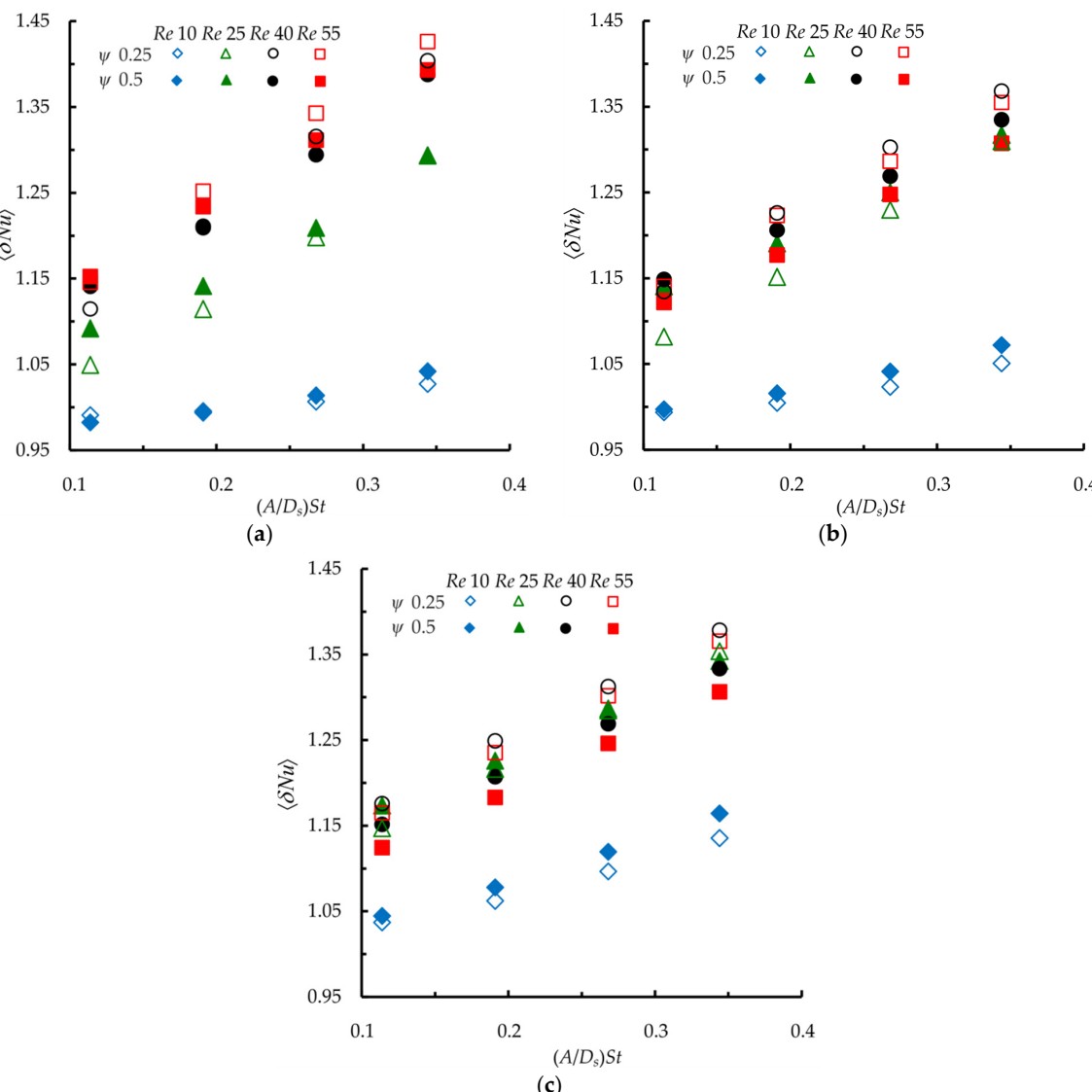

**Figure 13.** The effect of the pulsating flow on the Nusselt number ratio averaged over one period of pulsation for different porosities: (**a**) $\varepsilon$ = 0.743, (**b**) $\varepsilon$ = 0.864. (**c**) $\varepsilon$ = 0.954.

At a porosity of 0.864 (Figure 13b), as well as at a porosity of 0.743, the effect of the duty cycle depended on the Reynolds number. The symmetrical pulsations were more effective for heat transfer intensification at Reynolds numbers of 10 and 25, and for the asymmetric pulsations at Reynolds numbers of 40 and 55, except for regimes with a minimum intensity of pulsations. The difference between the symmetric and asymmetric pulsations increased with increasing pulsation intensity at all Reynolds numbers, except *Re* 25. When the Reynolds number *Re* was 25, the difference between symmetric and asymmetric pulsations decreased with increasing pulsation intensity. The ratio of the Nusselt number between the symmetric and asymmetric pulsation differed by 5.4%. For VF with a porosity of 0.864, were cases in which heat transfer enhancement was not observed. Decline of heat transfer enhancement was observed at the minimum Reynolds number and intensity of pulsation. With an increase in Reynolds numbers greater than 10, the heat transfer intensification increased significantly compared to *Re* less than 25. At *Re* 10 (Figure 13b), the heat transfer intensification increased by factor 1.07. The Nusselt number ratio for the same pulsation duty cycle increased with an increase at a Reynolds number of 40, and decreased insignificantly at *Re* 55. The maximum values of the Nusselt number ratio for the asymmetric pulsations at *Re* 40 and 55 were 1.37 and 1.36, respectively. For

symmetrical pulsations at *Re* 40 and 55, the heat transfer intensification increased by factor 1.33 and 1.31, respectively.

When the porosity was 0.954 (Figure 13c), the effect of pulsating flow on heat transfer was similar to porosities of 0.743 and 0.864. At Reynolds numbers of 40 and 55, the Nusselt ratio was generally higher for the asymmetric pulsations. When the Reynolds number was 25, the duty cycle effect of the pulsations depended on the intensity of the pulsations. For *Re* 25, at the minimum pulsation intensity, the Nusselt number ratio was higher for the symmetric pulsations; however, with an increase in $(A/D_s)St$ to 0.344, the heat transfer augmentation was higher for the asymmetric pulsations. The difference between the symmetric and asymmetric pulsations for the entire studied range at a porosity of 0.954 did not exceed 4.5%. At a porosity of 0.954, heat transfer intensification was observed for the entire studied range. For $(A/D_s)St$ of 0.114 and *Re* 10, the heat transfer augmentation was 1.8%. For symmetrical pulsations and the minimum intensity of pulsations, the maximum intensification of heat transfer was observed at *Re* 25, and for the asymmetric pulsations at *Re* 40. At higher values of $(A/D_s)St$, the enhancement of heat transfer was higher for pulsations with a duty cycle of 0.25 and *Re* 40. For asymmetric pulsations, the maximum intensification of the heat transfer was 1.38 at *Re* 40, and for symmetrical pulsations, the heat transfer enhancement was 1.34 at *Re* 25.

Figure 14 shows the effect of porosity on the heat transfer enhancement at $(A/D_s)St$ of 0.268 for various Reynolds numbers and duty cycles. Figure 14 shows that the effect of porosity on the Nusselt number ratio was different depending on the Reynolds number. For Reynolds numbers of 10 and 25, with an increase in porosity, an increase in the heat transfer enhancement occurred. When the Reynolds numbers were 40 and 55, with an increase in porosity from 0.743 to 0.864, the heat transfer intensification decreased, with a further increase in porosity, and the Nusselt number ratio did not change significantly. These trends were similar for both asymmetric (Figure 14a) and symmetrical pulsations (Figure 14b).

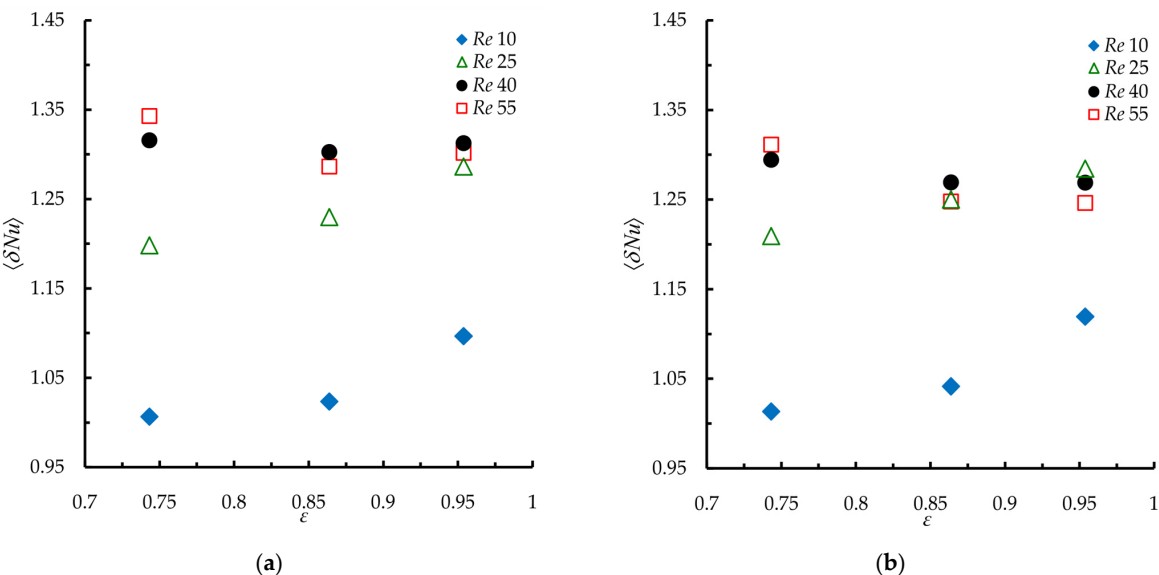

**Figure 14.** The effect of the porosity on the Nusselt number ratio averaged over one period of pulsation for different duty cycles: (**a**) $\psi = 0.25$, $(A/D_s)St = 0.268$. (**b**) $\psi = 0.5$, $(A/D_s)St = 0.268$.

### 3.3. The Effect of Pulsation on the Friction Factor Ratio

The influence of the pulsating flow on the friction factor was estimated from the friction factor ratio, which was calculated by the formula:

$$\langle \delta\zeta \rangle = \frac{\langle \zeta_p \rangle}{\zeta_{st}}, \tag{11}$$

where $\zeta_{st}$ is the friction factor for steady flow, which was calculated as follows:

$$\zeta_{st} = \frac{\Delta P2}{\rho u_{st}^2}.$$

Figure 15 shows the friction factor ratio obtained for various VF porosities. Figure 15 shows that for all the studied porosities, the higher the pulsation intensity, the higher the friction factor ratio, which was consistent with other works [17,18,58]. An increase in the friction factor was associated with an increase in instantaneous flow velocity compared to steady flow. The duty cycle of the pulsations had a significant impact on the friction factor ratio. For all porosities at the same Reynolds number, the friction factor ratio was always higher for symmetrical pulsations. For example, at a porosity of 0.743, *Re* 55 and $(A/D_s)St$ of 0.344 (Figure 15a), the friction factor ratio was 7.83 and 11.45 for asymmetrical and symmetrical pulsation, respectively.

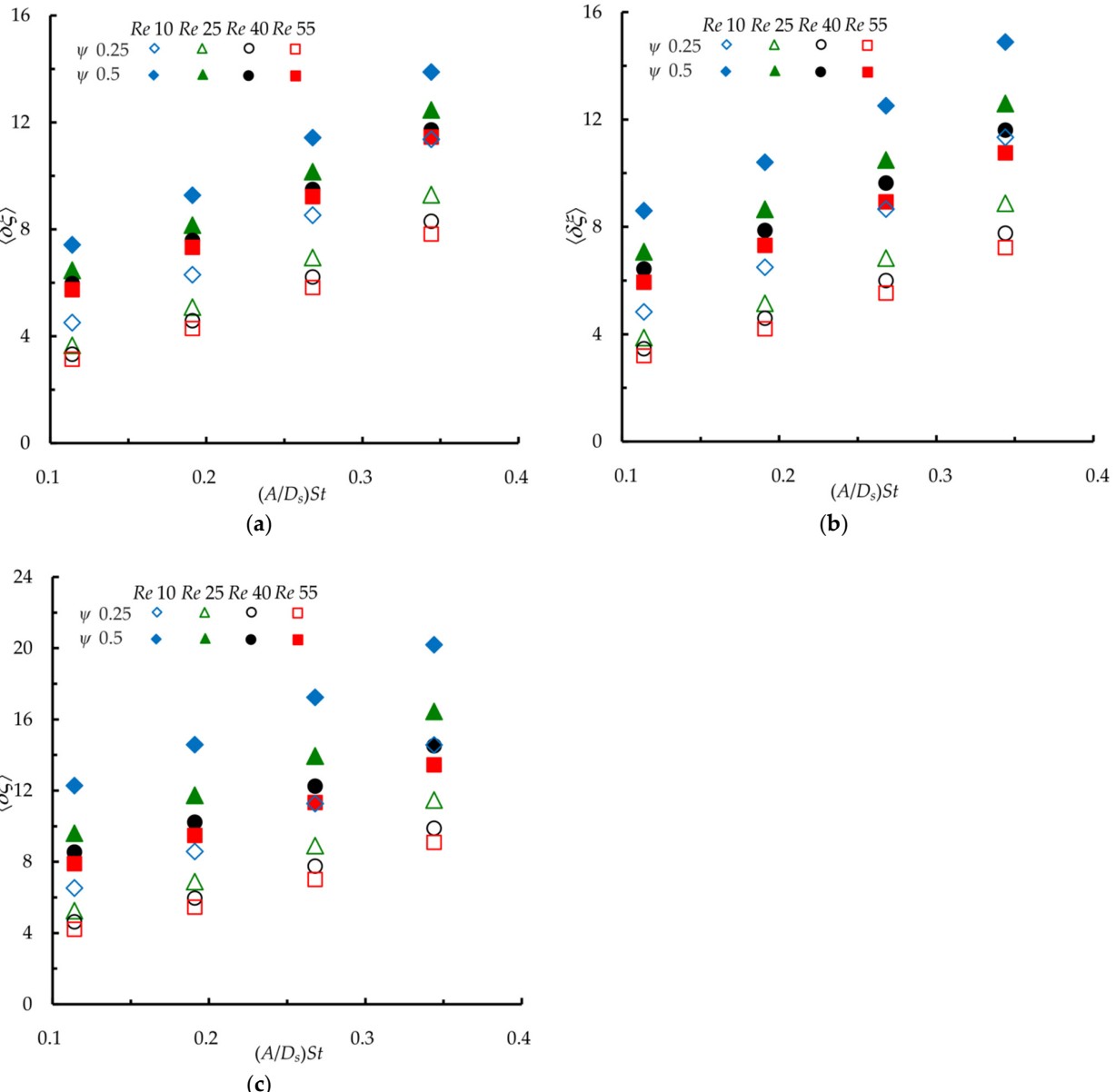

**Figure 15.** The effect of the pulsating flow on the friction factor ratio averaged over one period of pulsation for different porosities: (**a**) $\varepsilon = 0.743$. (**b**) $\varepsilon = 0.864$. (**c**) $\varepsilon = 0.954$.

When the porosity was 0.743 (Figure 15a), with an increase in the pulsation intensity, an increase in the friction factor ratio at all Reynolds numbers and pulsation duty cycles occurred. At the minimum Reynolds number, the friction factor increased from 7.41 to 13.89 and from 4.5 to 11.37 for symmetrical and asymmetrical pulsations, respectively. When the Reynolds number was at its maximum, the friction factor increased from 5.74 to 11.45 and from 3.14 to 7.83 for symmetric and asymmetric pulsations, respectively. For both symmetrical and asymmetric pulsations, as the Reynolds number decreased, the friction factor ratio increased. A more significant increase in the friction factor ratio was observed with a decrease in the Reynolds number from 25 to 10. For example, with a porosity of 0.743, a pulsation intensity of 0.114 and a duty cycle of 0.25 (Figure 15a), with a decrease in the Reynolds number from 55 to 25, the friction factor ratio was increased from 3.14 to 3.66. With a decrease in the Reynolds number to 10, the friction factor ratio increased to 4.5. With symmetric pulsations, the friction factor ratio was higher than asymmetric ones, regardless of the Reynolds numbers. Although the friction factor ratio increased with decreasing Reynolds numbers, for asymmetric pulsations and for the minimum Reynolds number, the friction factor ratio was lower than for symmetric pulsations with the maximum Reynolds number. For example, for asymmetric pulsations, the friction ratio was 4.5 at $Re$ 10 and $(A/D_s)St$ of 0.114, and with an increase in the Reynolds number to 55 for symmetrical pulsations, the friction ratio was 5.74. The maximum value of the friction ratio for asymmetrical pulsations was 11.37 at $Re$ 10 and $(A/D_s)St$ of 0.344. The minimum value of the friction ratio for asymmetrical pulsations was 3.14 at $Re$ 55 and $(A/D_s)St$ of 0.114. For symmetric pulsations, the friction ratio had a maximum value of 13.89, also at the minimum Reynolds number and maximum pulsation intensity. The minimum value of the friction ratio for symmetrical pulsations was 5.74 at $Re$ 55 and $(A/D_s)St$ of 0.114.

When the porosity was 0.864 (Figure 15b), the effect of the Reynolds number, pulsation intensity, and pulsation duty cycle on the friction ratio was similar to with a porosity of 0.743. With an increase in the Reynolds number for symmetric and asymmetric pulsations, a decrease in the friction ratio was observed. An increase in the intensity of pulsations has always been associated with an increase in the friction ratio. With symmetric pulsations, the friction ratio was significantly higher than with asymmetric pulsations. For example, with a Reynolds number of 10 and a pulsation intensity of 0.191, the friction ratio was 6.5 and 10.41 for asymmetric and symmetrical pulsations, respectively. The difference between symmetric and asymmetric pulsations at fixed Reynolds numbers increased with decreasing pulsation intensity. For example, with $Re$ 10 and $(A/D_s)St$ of 0.344, the difference between symmetric and asymmetric pulsations for the friction ratio was 31%, and with a decrease in $(A/D_s)St$ to 0.114, the difference was 78%. For asymmetric pulsations, the minimum friction factor ratio was 3.21, with $Re$ 55 and minimum pulsation intensity. The maximum friction ratio for asymmetrical pulsations was 11.33 at $Re$ 10 and maximum pulsation intensity. For symmetrical pulsations, the maximum friction factor ratio was 14.89 at $Re$ 10 and maximum pulsation intensity. The minimum value of the friction factor ratio for symmetrical pulsations was 5.94, which was observed at $Re$ 55 at the minimum pulsation intensity.

At a porosity of 0.954 (Figure 15c), as well as at porosities of 0.743, the effect of the Reynolds number, pulsation intensity, and pulsation duty cycle on the friction factor ratio is similar to other porosities. The friction factor ratio was always higher for symmetrical pulsations. However, with increasing porosity, the difference between symmetric and asymmetric pulsations increased at fixed Reynolds numbers. For example, with the minimum Reynolds number and the intensity of pulsations, the difference between symmetrical and asymmetric pulsations for the porosity of 0.954 was 88%, and for the porosities of 0.864 and 0.743, the differences were 78% and 65%, respectively. At a porosity of 0.954, the lowest friction factor ratio of 4.22 was observed for asymmetric pulsations at $Re$ 55 and $A/D_s)St$ of 0.114. The maximum friction factor ratio for asymmetrical pulsations was 14.57 at $Re$ 10 and $(A/D_s)St$ of 0.344. For the symmetrical pulsations, the maximum friction factor ratio was 20.2, which was observed at the minimum Reynolds number. The minimum value of

the friction factor ratio for the symmetrical pulsations was 7.9 at the maximum Reynolds number and minimum pulsation intensity.

Figure 16 shows the effect of porosity on the friction factor ratio for $(A/D_s)St$ of 0.268. Figure 16 shows that porosity had a slight effect on the friction factor ratio in the porosity range from 0.743 to 0.864. As the porosity increased from 0.864 to 0.954, the friction factor ratio increased for both duty cycles and all Reynolds numbers. Dellali et al. [21] also found that the increase in friction factor ratio in a pulsating flow compared to a steady flow was higher for the highest porosity.

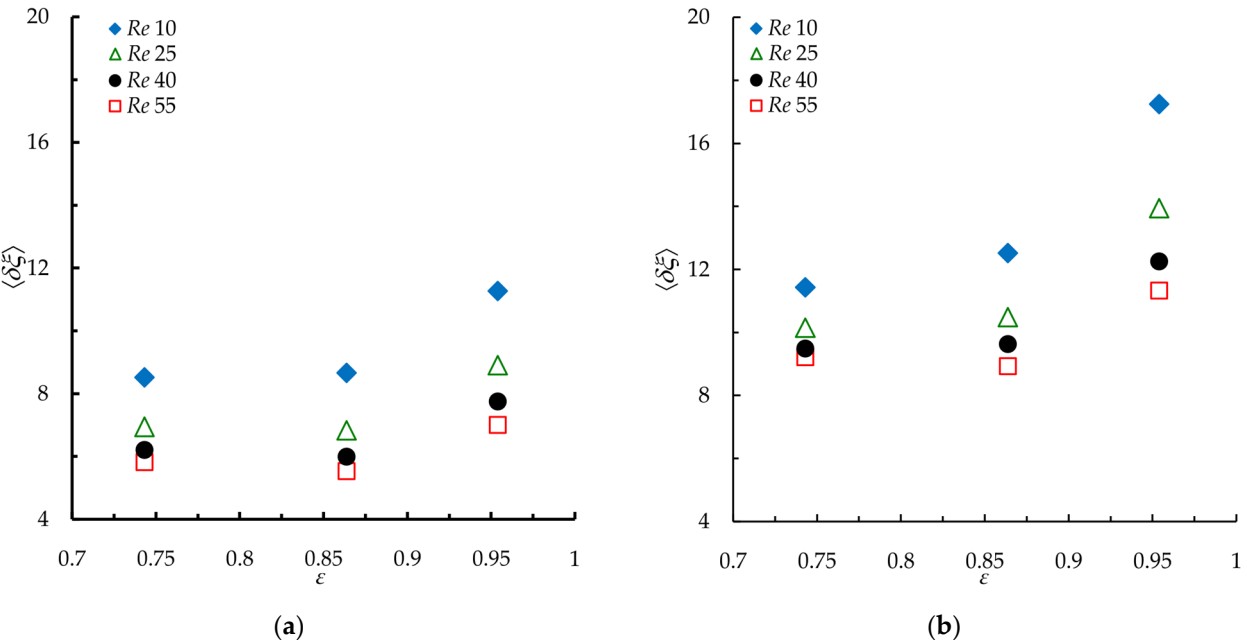

**Figure 16.** The porosity effect on the friction factor ratio averaged over one period of pulsation for different duty cycles: (**a**) $\psi = 0.25$, $(A/D_s)St = 0.268$. (**b**) $\psi = 0.5$, $(A/D_s)St = 0.268$.

*3.4. The Effect of Pulsations on the Thermal Performance Factor*

There are various methods for evaluating the effectiveness of heat transfer intensification methods [59]. One of these methods is to compare between heat transfer intensification and the friction factor increase at the same Reynolds number. The evaluation of the efficiency of pulsations for heat transfer intensification was performed with the Thermal Performance Factor (TPF) at the same Reynolds number. TPF was defined by the equation:

$$\text{TEF} = \frac{\langle \delta Nu \rangle}{\langle \delta \zeta \rangle}. \tag{12}$$

Figure 17 shows the TPF of the VF under pulsating flow conditions. For the entire studied range, TPF was less than one, which indicates that the increase in friction factor ratio in a pulsating flow is always higher than the increase in heat transfer intensification. An increase in the pulsation intensity leads to a decrease in TPF, which is consistent with the Haibullina et al. paper [60] on pulsating flow in tube bundles. The intensification of heat transfer can be higher both with the symmetric and asymmetric pulsations (Figure 13). However, TPF at the same Reynolds number was always higher for the asymmetric flow pulsations, because the friction factor ratio for symmetric pulsations was always higher than asymmetric pulsation (Figure 15). The friction factor ratio was higher at the lowest Reynolds number (Figure 15), while the Nusselt ratio was higher at the highest Reynolds number (Figure 13), so an increase in the Reynolds number leads to an increase in TPF. The highest TPF values for all studied porosities of 0.743, 0.864, and 0.954 were 0.37, 0.35, and

0.28, respectively. The maximum TPF value for all porosities was observed at asymmetric pulsation, minimal pulsation intensity and maximum Reynolds number.

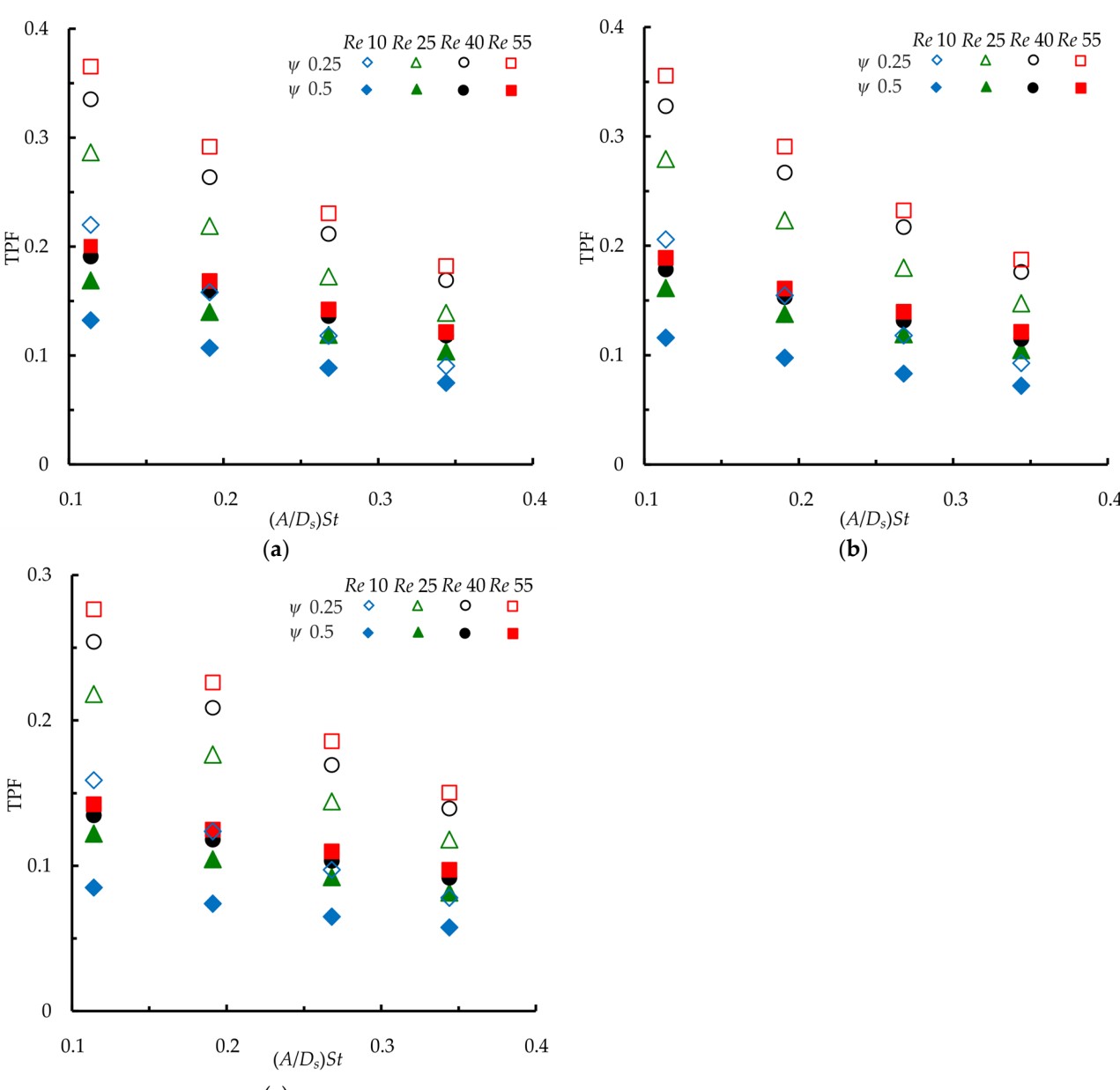

**Figure 17.** The effect of the pulsating flow on the Thermal Performance Factor averaged over one period of pulsation for different porosities: (**a**) $\varepsilon = 0.743$. (**b**) $\varepsilon = 0.864$. (**c**) $\varepsilon = 0.954$.

Figure 18 shows the effect of porosity on TPF for various Reynolds number and duty cycles. The porosity increase in range from 0.743 to 0.864 had small effect on TPF. As the porosity increased to 0.954, TPF decreased. A decrease in TPF with an increase in porosity greater than 0.864 was consistent with an increase in the friction factor ratio (Figure 16).

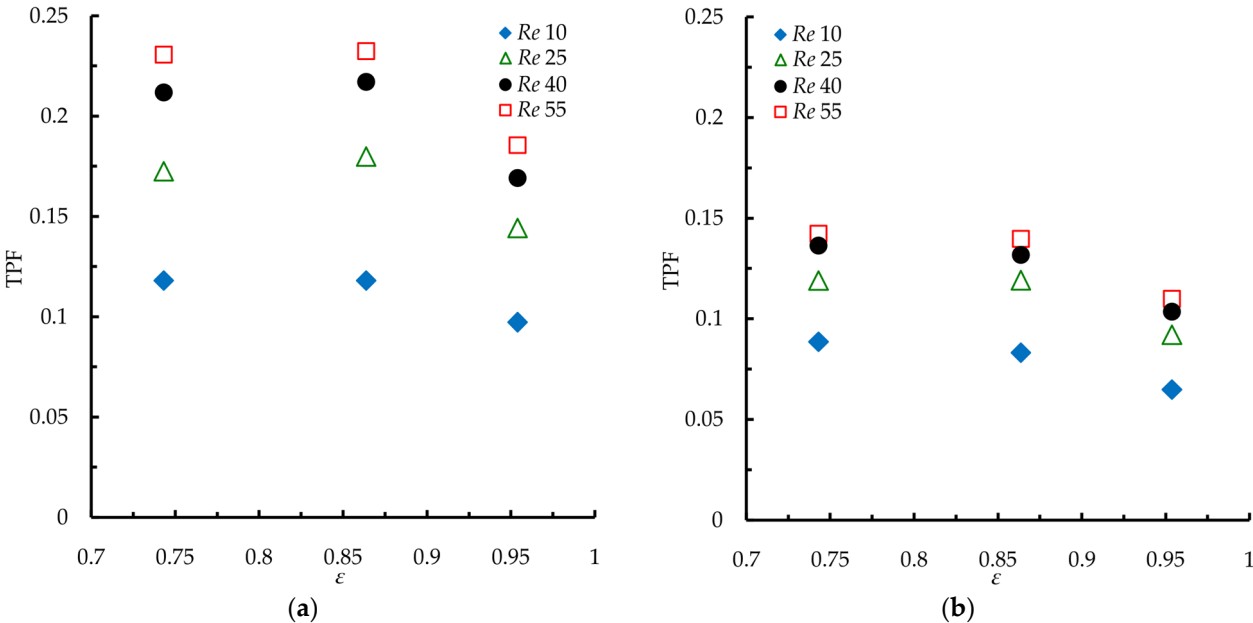

**Figure 18.** The effect of the porosity on the Thermal Performance Factor averaged over one period of pulsation for different duty cycles: (**a**) $\psi = 0.25$, $(A/D_s)St = 0.268$. (**b**) $\psi = 0.5$, $(A/D_s)St = 0.268$.

### 3.5. The Empirical Correlation for Predicting the Heat Transfer Enhancement, Friction Factor Ratio and TPF

According to the results of a numerical study, the empirical correlations were obtained to predict the Nusselt number ratio, friction factor ratio and TPF in VF, with asymmetric and symmetric pulsations. The Nusselt number ratio, friction factor ratio, and TPF with asymmetric pulsations can be calculated by Equations (13)–(15), respectively.

$$\langle \delta Nu \rangle = \frac{\langle Nu_p \rangle}{Nu_{st}} = 0.998Re^{0.125}((A/D_s)St)^{0.14}\varepsilon^{0.134}. \tag{13}$$

$$\langle \delta\xi \rangle = \frac{\langle \xi_p \rangle}{\xi_{st}} = 54.3Re^{-0.249}((A/D_s)St)^{0.755}\varepsilon^{0.954}. \tag{14}$$

$$TPF = 0.018Re^{0.374}((A/D_s)St)^{-0.615}\varepsilon^{-0.829}. \tag{15}$$

For symmetric pulsations, the Nusselt number ratio, friction factor ratio, and TPF at can be predicted by Equations (16)–(18), respectively.

$$\langle \delta Nu \rangle = \frac{\langle Nu_p \rangle}{Nu_{st}} = 1.038Re^{0.102}((A/D_s)St)^{0.124}\varepsilon^{0.082}. \tag{16}$$

$$\langle \delta\xi \rangle = \frac{\langle \xi_p \rangle}{\xi_{st}} = 53.1Re^{\cdot-0.195}((A/D_s)St)^{0.528}\varepsilon^{1.211}. \tag{17}$$

$$TPF = 0.02Re^{\cdot0.297}((A/D_s)St)^{-0.403}\varepsilon^{-1.129}. \tag{18}$$

The Reynolds number and the Strouhal number in Equations (16)–(18) were based on the struts' diameter. Equations (16)–(18) were obtained for the range of Reynolds numbers $10 \leq Re \leq 55$, dimensionless group $0.114 \leq (A/D_s)St \leq 0.344$, and porosities $0.743 \leq \varepsilon \leq 0.954$. Figure 19 compares the Nusselt number ratio, friction factor ratio, and TPF, predicted by Equations (13)–(18) and obtained from the numerical simulation. The ability of Equations (13)–(18) to predict the Nusselt number ratio, friction factor ratio, and TPF of the numerical simulation is in a satisfactory range of 20%. Equations for the Nusselt number ratio are in better agreement with numerical simulation data than the equations for the friction factor ratio and TPF. For the Nusselt number ratio, the maximum deviation was

7.5%. The maximum deviation for the friction factor ratio and TPF were 20% and 17.8%, respectively. The average deviation for the friction factor and TPF ratio were 7.5% and 6.8%, respectively.

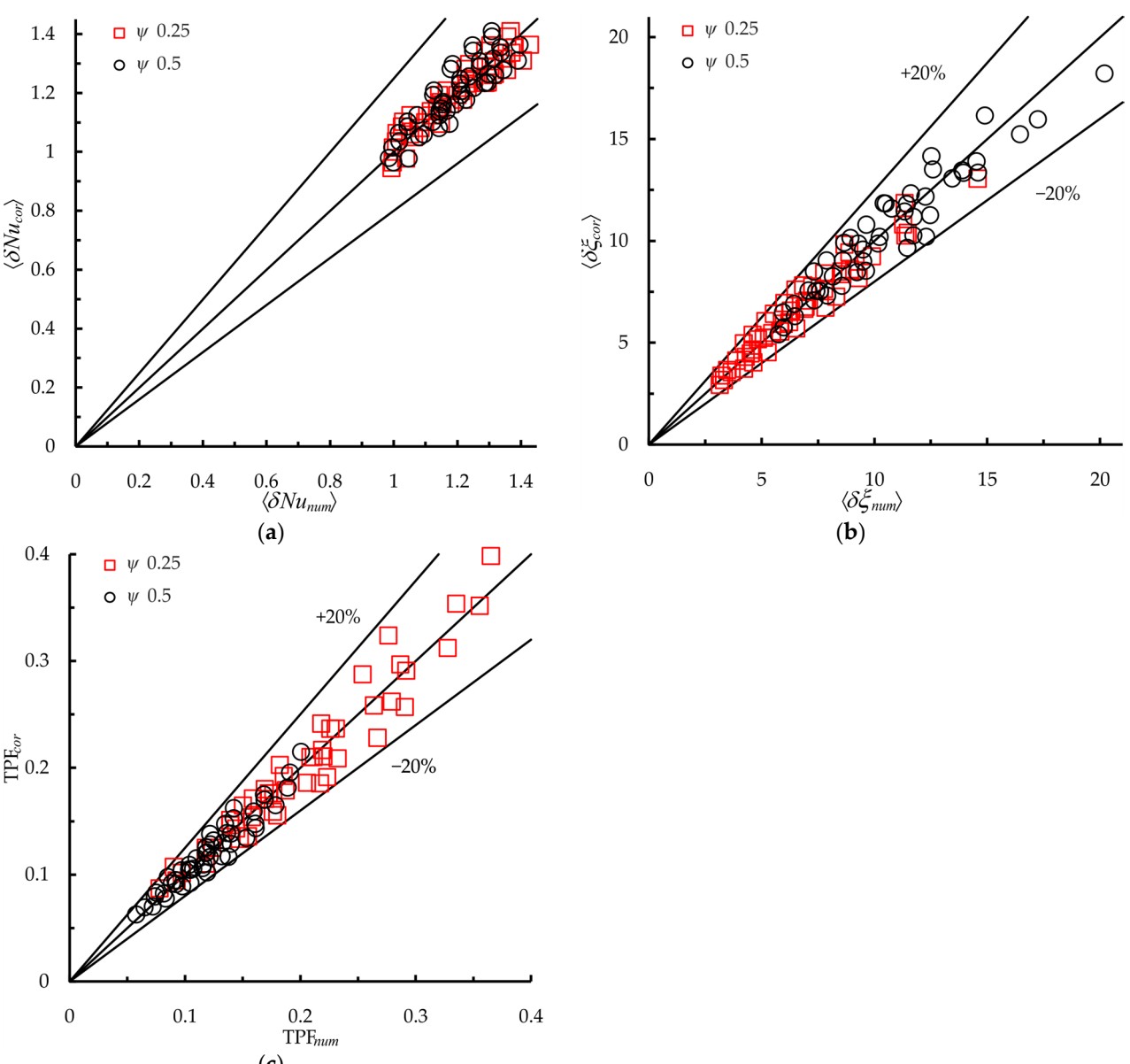

**Figure 19.** Comparison between correlations (13)–(18) and numerical simulation: (**a**) the Nusselt number ratio. (**b**) The friction factor ratio. (**c**) TPE.

The Nusselt number for steady flow in Equations (13) and (16) can be calculated from the following correlation:

$$Nu_{st} = 0.81 Re^{0.327} \varepsilon^{-1.668}. \tag{19}$$

Equation (15) was obtained from numerical simulation data for a steady flow and can be used for the range of Reynolds numbers $10 \leq Re \leq 55$ and porosities $0.743 \leq \varepsilon \leq 0.954$. The absolute deviation of Equation (15) with numerical simulation data was 8.51%. The interstitial heat transfer coefficient for steady flow can be calculated as follows:

$$h_{st} = \frac{Nu_{st}k}{D_s}.$$

## 4. Conclusions

In this work, the possibility of heat transfer enhancement in VF under pulsating flow with unidirectional flow conditions was studied using numerical simulation. The heat transfer enhancement was evaluated for both asymmetric and symmetric pulsations.

The results of the numerical study showed that the duty cycle effect on the pulsations was different depending on the Reynolds numbers. Symmetrical fluctuations showed slightly better heat transfer enhancement, by factor 4–5%, at low Reynolds numbers in the investigated range. Asymmetric pulsations, on the contrary, were more effective at high Reynolds numbers. The difference in the effect of symmetric and asymmetric pulsations was insignificant compared to the magnitude of the heat transfer intensification. The maximum difference in the heat transfer intensification between symmetric and asymmetric pulsations was 5.4%. The maximum heat transfer enhancement of 43% was achieved with the duty cycle of 0.25, a dimensionless group of 0.344, a Reynolds number of 55, and a porosity of 0.743. An increase in the pulsation intensity led to an increase in the heat transfer enhancement, regardless of the porosity and condition parameters. An increase in Reynolds numbers also led to an increase in the heat transfer intensification.

The TPF of the pulsation method of heat transfer enhancement in open-cell foams depends significantly on the pulsations' parameters. The TPF of pulsating flows decreased with increasing pulsation intensity and decreasing Reynolds numbers. Asymmetric pulsations had higher TPF values due to a higher increase in friction factor with symmetrical pulsations. VF with lower porosity had better TPF. The maximum TPF was 0.36, with a minimum pulsation intensity, a maximum Reynolds number, a pulsation duty cycle of 0.25, and porosity of 0.743.

According to the numerical study, empirical correlations were obtained to predict the convective heat transfer augmentation, friction factor ratio, and TPF under conditions of pulsating flow in open-cell foams.

**Supplementary Materials:** The following supporting information can be downloaded at: https://www.mdpi.com/article/10.3390/en15228660/s1, Video S1: Pulsating flow in the Voronoi foam.

**Author Contributions:** Conceptualization, A.K., A.H. and V.B.; methodology, A.S., L.K., D.B. and V.I.; software, A.K. and A.S.; validation, V.B. and L.K.; investigation, A.K. and A.H.; resources, V.B. and V.I.; formal analysis, V.B., L.K., D.B. and V.I.; writing—original draft preparation, A.K., A.H. and A.S.; writing—review and editing, A.K., A.H. and A.S.; visualization, A.K. and D.B.; supervision, A.H.; Project administration, A.K. All authors have read and agreed to the published version of the manuscript.

**Funding:** The research was funded by the Russian Science Foundation, grant number 21-79-10406, https://rscf.ru/en/project/21-79-10406/ accessed on 28 October 2022.

**Data Availability Statement:** The data presented in this study are available on request from the corresponding author.

**Conflicts of Interest:** The authors declare no conflict of interest.

## Abbreviations

| | |
|---|---|
| $A$ | Dimensional amplitude of pulsation [m] |
| $A/D_s$ | Dimensionless relative amplitude of pulsation [–] |
| $a_{sv}$ | Surface area per unit of volume [m$^{-2}$ m$^{-3}$] |
| $D_s$ | Strut diameter [m] |
| $f$ | Frequency of pulsation [Hz] |
| $h$ | Interstitial heat transfer coefficient [W m$^{-2}$ K$^{-1}$] |
| $k$ | Thermal conductivity [W m$^{-1}$ K$^{-1}$] |
| $Nu$ | Nusselt number based on strut diameter [–] |
| $Pr$ | Prandtl number |
| PPI | Pore number per linear inch (pores in$^{-1}$) |

| | |
|---|---|
| $q$ | Heat flux [W m$^{-2}$] |
| $Re$ | Reynolds numbers based on strut diameter [–] |
| $St$ | Strouhal number based on strut diameter [–] |
| $t_{air}$ | Mean air temperature [K] |
| $t_w$ | Mean wall temperature [K] |
| $u$ | Inlet velocity [m s$^{-1}$] |
| $y_{max}/D_s$ | Maximum mesh size related to the cell diameter [–] |
| $T$ | Period of the pulsation [s] |
| $T_1$ | First half-period of the pulsation [s] |
| $T_2$ | Second half-period of the pulsation [s] |

Greek symbols

| | |
|---|---|
| $\mu$ | Dynamic viscosity [Pa s] |
| $\rho$ | Density [kg m$^3$] |
| $\psi$ | Duty cycle [–] |
| $\varepsilon$ | Porosity [–] |
| $\xi$ | Friction factor [–] |
| $\Delta P$ | Pressure drop [Pa] |
| $\tau$ | Time [s] |

Subscripts

| | |
|---|---|
| $\delta$ | Enhancement factor |
| $cor$ | Correlation |
| $num$ | Numerical |
| $p$ | Pulsating flow |
| $st$ | Steady flow |

Abbreviations

| | |
|---|---|
| LVT | Laguerre–Voronoi tessellation |
| TPF | Thermal Performance Factor |
| VF | Voronoi foam |

Notations

| | |
|---|---|
| $\langle\rangle$ | Averaged value over one period of the pulsation |

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
