# Peer review of "Heat Transfer in 3D Laguerre–Voronoi Open-Cell Foams under Pulsating Flow"

_energies, doi:10.3390/en15228660_

Round 1

Reviewer 1 Report

The manuscript presents a numerical simulation of heat transfer in porous media under conditions of symmetric and asymmetric pulsations of fluid.  The Laguerre-Voronoi tessellation method was used to generate the 3D open cell foams and three foam porosities were analyzed.

The theme of the manuscript is part of the topics covered by Energies and I believe it is of great interest to journal’s readers.

The manuscript is well structured and referenced.

The 3D foam structure generation, the governing equation, the boundary condition and the methodology are explained in detail so that other researchers can repeat the simulation.

The grid independency was tested and the mathematical model was verified by comparing the simulation results with experimental ones, for a certain porosity.

The authors determined the influence of pulsations on Nusselt number, friction factor and thermal performance factor and, based on the results, empirical correlations were obtained to predict the heat transfer intensification in porous media for a steady and pulsating flow.

The results are clearly presented and they support the conclusions.

Author Response

Dear reviewer, thank you for your positive review report.

Reviewer 2 Report

In this paper, the authors studied the heat transfer capacity in open cell foams under pulsating flow with 3d laguerre-voronoi tessellation. The research topic is meaningful. The main problems in this paper are as follows:

Comments:

(1) Please describe the selection basis of condition parameters in this study. What equipment does this parameter range apply to?

(2) Does the grid model have a boundary layer?

(3) Please complete the flow direction identification for Figures 7 to 12, and add variables and units to the legend.

(4) Please supplement the quantitative analysis results of friction coefficient, similar to Figure 13.

(5) Please supplement the fitting correlations of friction coefficient and thermal performance factor.

(6) Where are equations (13) and (14)?

(7) The conclusion should be concise.

(8) Please check and revise the English writing carefully.

Author Response

Dear reviewer, thank you for your valuable comments and suggestion.

(1) We added the next lines to Results and discussion section before section 3.1.: The condition parameters of Reynolds and Prandtl number in this study were chosen to apply for applications in electronic cooling. The pulsating condition parameters were varied to investigate the effect on thermal performance VF. The maximum values of the dimensionless group (A/Ds)St were chosen to consider the cost of numerical simulation. Since some studies have shown the effectiveness of asymmetrical pulsations compared with symmetrical ones [50,51], both types of pulsations are considered in the study.

(2) We added the next lines to section 2.4.: The boundary layer has contained six layers. The cell size of the first three layers expanded in the radial direction with a factor of 1.2. 

(3) We completed the flow direction identification for Figures 7 to 12 and added variables and units to the legend.

(4) The quantitative analysis results of the friction coefficient have been added in section 3.3.

(5) The fitting correlations of friction coefficient and thermal performance factor were added in the paper.

(6) The authors would like to thank the reviewer for this suggestion especially. The equations (13), (14) were missing by mistake in the last version of the paper. We added equations (13) and (14) in section 3.5.

(7) The conclusion has been revised.

(8) We check and revised the English writing.

Reviewer 3 Report

Dear Editor/ Authors, 

           The topic is interesting, the work structure and the scientific content of the paper is good, the academic level of the paper is good, the conclusions are justified. Overall, this paper is of a good quality and well-written, the recommendation is to accept this paper for publication.

Best Regards

Author Response

(The authors gave the same response as above.)
